# GATEKEEPER: Improving Model Cascades Through Confidence Tuning

**Stephan Rabanser**[*]
Princeton University
rabanser@princeton.edu

**Nathalie Rauschmayr**
Google
rauschmayr@google.com

**Achin Kulshrestha**
Google
kulac@google.com

**Petra Poklukar**
Google
poklukar@google.com

**Wittawat Jitkrittum**
Google
wittawat@google.com

**Sean Augenstein**
Google
saugenst@google.com

**Congchao Wang**
Google
congchaowang@google.com

**Federico Tombari**
Google
tombari@google.com

## Abstract

Large-scale machine learning models deliver strong performance across a wide range of tasks but come with significant computational and resource constraints. To mitigate these challenges, local smaller models are often deployed alongside larger models, relying on routing and deferral mechanisms to offload complex tasks. However, existing approaches inadequately balance the capabilities of these models, often resulting in unnecessary deferrals or sub-optimal resource usage. In this work we introduce a novel loss function called GATEKEEPER for calibrating smaller models in cascade setups. Our approach fine-tunes the smaller model to confidently handle tasks it can perform correctly while deferring complex tasks to the larger model. Moreover, it incorporates a mechanism for managing the trade-off between model performance and deferral accuracy, and is broadly applicable across various tasks and domains without any architectural changes. We evaluate our method on encoder-only, decoder-only, and encoder-decoder architectures. Experiments across image classification, language modeling, and vision-language tasks show that our approach substantially improves deferral performance.

## 1 Introduction

In recent years, large-scale machine learning models such as Gemini (GeminiTeam et al., 2023), GPT-4 (Achiam et al., 2023) or Claude (Anthropic, 2024) have gained significant traction due to their remarkable ability to address a wide array of tasks. These tasks range from natural language understanding and generation, including machine translation, summarization, and conversational agents, to computer vision applications like image recognition, object detection, and image captioning. The versatility and high performance of these expansive models make them invaluable tools across diverse domains, including healthcare (Nazi & Peng, 2024), finance (Li et al., 2023), education (Wang et al., 2024b), and entertainment (Gallotta et al., 2024).

Deploying and operating such large models presents significant challenges in terms of latency, memory, compute and storage (Pope et al., 2023). Optimizing inference costs is an active research

---

[*]Work done while a Student Researcher at Google.

39th Conference on Neural Information Processing Systems (NeurIPS 2025).

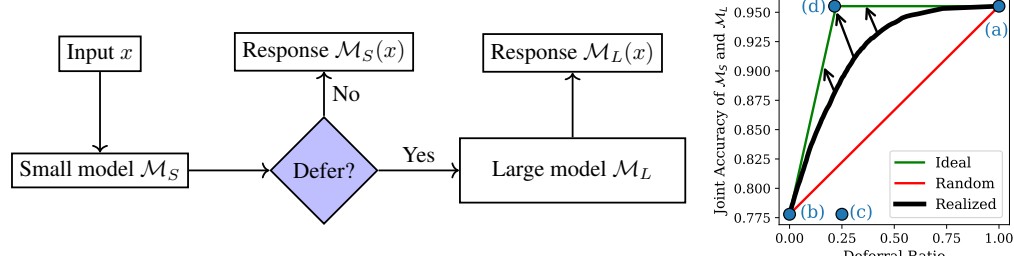

Figure 1: **Overview of the cascading setup (left) and performance trade-off (right)**. *Left*: Cascading determines which inputs should be predicted by a small model $\mathcal{M}_S$ or routed to a large model $\mathcal{M}_L$. *Right*: Performance is measured as a trade-off between joint accuracy across $\mathcal{M}_S$ and $\mathcal{M}_L$ and deferral ratio. Ideal deferral strategies optimize this trade-off and push the realized deferral curve closer to the ideal deferral depicted in (d). (a) depicts full deferral; (b) depicts no deferral; and (c) depicts excessive deferral of requests that could have been correctly handled by $\mathcal{M}_S$.

area which includes both techniques for reducing the size of the existing large model such as model compression (Hoefler et al., 2021), model pruning (Ma et al., 2023; Cheng et al., 2024) and distillation (Yang et al., 2024), and those aiming to leverage a sequence of models such as speculative decoding (Leviathan et al., 2023) and model cascades (Dohan et al., 2022; Chen et al., 2023a; Gupta et al., 2024; Chen et al., 2024a). However, due to scaling laws showing that the performance of a Large Language Model (LLM) increases with its size (Kaplan et al., 2020), the latter category of methods leveraging a sequence of models is currently a more promising direction to lower inference costs without sacrificing the capabilities of large models.

Both speculative decoding and model cascading rely on the existence of a large performant model $\mathcal{M}_L$ and a small model $\mathcal{M}_S$ that is cheap, fast, and less accurate. Speculative decoding leverages $\mathcal{M}_S$ for generating a set of draft tokens that are then validated by $\mathcal{M}_L$ in parallel, a technique successfully deployed in industry applications (Leviathan, 2024). In contrast, model cascades leverage a deferral rule for selecting the most suitable model to process a given request (see Figure 1 left). While the success of speculative decoding necessitates a highly performant $\mathcal{M}_S$ to generate quality draft tokens, model cascades allow the deployment of a less capable $\mathcal{M}_S$ by invoking $\mathcal{M}_L$ only for inference requests outside the small model's scope. In this work, we contribute to the advancement of the model cascades.

Model cascades achieve efficient deferral by optimizing two objectives: compute budget and joint accuracy. We illustrate this trade-off in Figure 1 (right). Assume we have *x* inference requests and a small model $\mathcal{M}_S$ that uses only *20%* of the compute budget of the large model $\mathcal{M}_L$. There are three worst-case scenarios: (a) $\mathcal{M}_S$ defers all requests to $\mathcal{M}_L$, yielding the highest joint accuracy (equal to that of $\mathcal{M}_L$) but the worst compute budget (*1.2x*), since both models process all inputs; (b) $\mathcal{M}_S$ never defers, achieving the lowest compute budget (*0.2x*) but also the lowest joint accuracy (equal to that of $\mathcal{M}_S$); (c) $\mathcal{M}_S$ defers only requests it would have answered correctly, increasing compute budget over (b) without improving joint accuracy. In contrast, the ideal case (d) occurs when $\mathcal{M}_S$ defers only the requests it would misclassify, achieving optimal joint accuracy for a given compute budget (*0.2–1x*). We define how closely a model approximates this ideal as its *deferral performance*.

In this paper, we address the following research question:

> *How can we optimize model cascades to maximize deferral performance?*

In other words, we focus on designing effective model cascades by making the small model more aware of what it does not know. We do so by introducing a *general-purpose* loss function, called GATEKEEPER, that calibrates the small model's confidence in its predictions. By fine-tuning $\mathcal{M}_S$ to output high confidence for correct predictions and low confidence for incorrect ones, we improve the reliability of its uncertainty estimates and facilitate learning of common tasks—thereby directly improving deferral performance. Crucially, GATEKEEPER includes a built-in mechanism for managing the trade-off between *model* and *deferral* performance, and is applicable to arbitrary architectures.

We empirically demonstrate the efficacy of the GATEKEEPER loss on encoder-only vision models for image classification, decoder-only language models (LMs) for closed-form text generation, and encoder-decoder models for vision-language (VL) tasks such as open set classification and captioning. Our main results show that models trained with GATEKEEPER outperform an untuned baseline by 0.72x/2x on CIFAR-100/TinyImagenet and 7x/10x on ARC-e/c, respectively, in terms of deferral performance. As a result, GATEKEEPER paves the way for more scalable and efficient deployment strategies, leveraging the collaboration between local and large-scale models to deliver high-quality results in applications with real-time processing demands.

## 2 Related Work

Our proposed method improves model cascades through uncertainty-aware finetuning. Next, we describe related work for both research areas. We provide an extended discussion in Appendix B.1.

**Model Cascades.** A cascade consists of a series of models and a deferral rule which determines the appropriate model given an input request. The concept of model cascades has first been proposed by Viola & Jones (2001), where it is used to accelerate object detection models. Cascades have been extensively studied for classification-based computer vision (Wang et al., 2017; Trapeznikov & Saligrama, 2013; Bolukbasi et al., 2017a; Jitkrittum et al., 2023) and in models for natural language processing (Dohan et al., 2022; Mamou et al., 2022; Varshney & Baral, 2022).

Cascades are particularly promising in the context of generative models such as LLMs and VLMs since they can significantly reduce inference costs. In contrast to speculative decoding (Leviathan et al., 2023), they aim to invoke the large model only for difficult examples. However, the two approached can also be combined (Narasimhan et al., 2025). While Chen et al. (2024b) combine the deferral logic with speculative decoding to generate initial tokens using larger models and later tokens using a smaller model, the majority of research on model cascades has focused on using pre-trained LLMs with a post-hoc deferral logic (Narasimhan et al., 2022; Jitkrittum et al., 2023; Yue et al., 2024). Kolawole et al. (2024) use agreement across multiple models to make deferral decisions, while Gupta et al. (2024) present a method to learn a deferral rule based on quantiles of per-token log probabilities.

Model cascades can be improved through training and fine-tuning. Wang et al. (2024a) train the small model only on easier examples by masking tokens where both large and small models are incorrect. Enomoro & Eda (2021) extend the training objective of image classification models with confidence calibration. In contrast, our approach extends cascades to VLMs and boosts inference performance by making smaller models less confident when incorrect.

**Uncertainty-Aware Models.** Extensive research has been conducted in the field of uncertainty quantification in deep learning and we refer to Abdar et al. (2021) for a detailed survey. While many methods have been proposed for classification-based models, measuring uncertainty for generative models is still an active area of research. Based on the assumed level of access to model internals, existing methods can be summarized into three main categories:

*Black box* methods operate solely via the model's query interface by injecting tailored instructions into prompts. These modify the prompt $\mathbf{x}$ by appending instructions $\mathbf{x}'$ for the model to respond less confidently: $\mathbf{x} \leftarrow \mathbf{x}|\mathbf{x}'$. Related methods are confidence quantification (Shrivastava et al., 2023), rejection and remote model awareness (Kadavath et al., 2022), and self-critiquing (Gou et al., 2023). Xiong et al. (2024) show that LLMs can express their confidence through prompting and sampling strategies and their experiments indicate that these models tend to be overconfident.

*Gray box* approaches employ confidence-based strategies centered on post-processing the model's logits. Many uncertainty techniques such as ensembling (Lakshminarayanan et al., 2017) and Bayesian methods (Blundell et al., 2015)) are not scalable. Related techniques are max confidence (Hendrycks & Gimpel, 2016), predictive entropy, and confidence reduction prompting. Malinin & Gales (2021) uses token-entropy as a measure of uncertainty in auto-regressive models and Kuhn et al. (2023) leverages linguistic invariances via semantic entropy.

*White box* methods use uncertainty-aware fine-tuning to produce better-calibrated models. Chuang et al. (2024) introduces Self-REF, a framework that leverages confidence tokens during fine-tuning to improve downstream routing. Krishnan et al. (2024) proposes an uncertainty-aware causal language modeling loss that captures the trade-off between accuracy and calibration. In contrast, our method calibrates the model so that correct predictions receive low uncertainty and incorrect ones high

uncertainty. We apply this uncertainty-aware model in a cascade inference system, where it improves overall performance. Prior work by Rawat et al. (2021) pre-partitions data into easy and hard examples, e.g., based on $\mathcal{M}_L$'s confidence, and trains $\mathcal{M}_S$ with explicit labels. We improve on this static partitioning by dynamically assigning examples during training based on $\mathcal{M}_S$'s current state.

# 3 The GATEKEEPER Loss

## 3.1 Overview & Setup

Our framework consists of a large, highly capable model $\mathcal{M}_L$ and a smaller, resource-efficient model $\mathcal{M}_S$. We assume that $S \in \mathbb{N}$ and $L \in \mathbb{N}$ represent the parameter count of each model with $S \ll L$. Both models can either function as classifiers (i.e., $\mathcal{M} : \mathbb{R}^D \rightarrow [C]$ with $\mathbb{R}^D$ denoting the input space and $C$ the number of total classes), or (multi-modal) sequence models (i.e., $\mathcal{M} : \mathbb{R}^D \rightarrow [V]^T$ where $V$ is the vocabulary and $T$ is the sequence length). We include experiments on all of these model classes in Section 4. Furthermore, we do not require a shared model family to be deployed on both $\mathcal{M}_S$ and $\mathcal{M}_L$; for example, $\mathcal{M}_S$ could be a custom convolutional neural network optimized for efficient inference and $\mathcal{M}_L$ a vision transformer (Dosovitskiy, 2020). The primary objective is to design a deferral mechanism that enables $\mathcal{M}_S$ to decide when to return its predictions without the assistance of $\mathcal{M}_L$ and when to instead defer to it. We assume that $\mathcal{M}_L$ is either outside of our control (e.g., an API endpoint) or too costly to modify, and that only $\mathcal{M}_S$ is subject to adaptation.

Deferral decisions are made using signals derived from the small model $\mathcal{M}_S$ as this approach is typically more cost-effective than employing a separate routing mechanism (Teerapittayanon et al., 2016). Approaches that involve querying the large model $\mathcal{M}_L$ to assist in making deferral decisions at test time are excluded from our setup. Such methods—common in domains like LLMs—are counterproductive to our goal since querying $\mathcal{M}_L$ defeats the purpose of making a deferral decision in the first place. Examples of these inapplicable methods include collaborative LLM frameworks (Mielke et al., 2022) and techniques that rely on semantic entropy for uncertainty estimation (Kuhn et al., 2023). As part of our setup, we assume that $\mathcal{M}_L$ dominates $\mathcal{M}_S$ as per the following assumption.

**Dominance Assumption.** Let $\mathcal{D}$ denote the target deployment distribution defined over covariates $\mathcal{X}$ and labels $\mathcal{Y}$. We assume that $\mathcal{M}_L$ dominates the $\mathcal{M}_S$ with high probability under $\mathcal{D}$; formally,

$$\Pr_{(x,y)\sim\mathcal{D}}\left[\mathcal{M}_L(x) \neq y \land \mathcal{M}_S(x) = y\right] \leq \delta, \tag{1}$$

with $\delta \ll 1$. This "almost-always" dominance, supported by scaling-law trends (Kaplan et al., 2020), implies that deferring from $\mathcal{M}_S$ to $\mathcal{M}_L$ cannot hurt accuracy in expectation, while still allowing rare counter-examples where the small model outperforms the large model. Note that we empirically observe $\delta = 0$ across all tasks considered in this work, meaning that $\mathcal{M}_L$ strictly dominates $\mathcal{M}_S$.

As discussed in Section 2, the choice of deferral strategy often depends on the level of access available to $\mathcal{M}_S$. We assume white box access with full access to $\mathcal{M}_S$'s internals. As such, deferral mechanisms can be directly integrated into the model's architecture and parameters. This involves fine-tuning $\mathcal{M}_S$ to predict deferral decisions or to incorporate rejection mechanisms within its predictive process. Our work falls into this category as it proposes a new loss function to fine-tune $\mathcal{M}_S$.

Our goal is to train a small model that can effectively distinguish between correct and incorrect predictions. While many past works have considered the question of whether it is possible to find proxy measures for prediction correctness, the central question we ask is:

*Can we optimize the small model to separate correct from incorrect predictions?*

We show that this is indeed achievable through a carefully designed fine-tuning stage that does not require any architectural modifications. This ensures that the ability to separate correct from incorrect decisions is integrated seamlessly into $\mathcal{M}_S$'s existing structure.

## 3.2 Confidence-Tuning for Deferral

**Prerequisite: Standard Training.** We begin with an $\mathcal{M}_S$ that has already been trained on the tasks it is intended to perform upon deployment. However, due to its limited capacity, $\mathcal{M}_S$ cannot achieve

the performance levels of $\mathcal{M}_L$. Importantly, we make no assumptions about the training process of $\mathcal{M}_S$ —whether it was trained from scratch without supervision from an external model or with the help of soft labels through a distillation approach.

**Stage 1 (Finetuning): Correctness-Aware Finetuning with GATEKEEPER.** Next, we introduce a correctness-aware loss, dubbed GATEKEEPER, to fine-tune $\mathcal{M}_S$ for improved confidence calibration. Specifically, the model is trained to make correct predictions with high confidence while reducing the confidence of incorrect predictions (see Figure 2). This loss can either rely on true labels or utilize the outputs of $\mathcal{M}_L$ with soft probabilities as targets.

In its canonical form, GATEKEEPER is defined as a hybrid loss $\mathcal{L} = \alpha\mathcal{L}_{\text{corr}} + (1 - \alpha)\mathcal{L}_{\text{incorr}}$ with

$$\mathcal{L}_{\text{corr}} = \frac{1}{N}\sum_{i=1}^{N}\mathbb{1}\{y_i = \hat{y}_i\}\text{CE}(p_i(\mathbf{x}_i), y_i) \qquad \mathcal{L}_{\text{incorr}} = \frac{1}{N}\sum_{i=1}^{N}\mathbb{1}\{y_i \neq \hat{y}_i\}\text{KL}\left(p_i(\mathbf{x}_i) \,\|\, \mathcal{U}\right). \quad (2)$$

Here, $y_i$ and $\hat{y}_i$ are the true and predicted labels for $\mathbf{x}_i$, respectively, $p_i$ is the predicted probability distribution of $\mathcal{M}_S$ over classes, $\mathcal{U}$ represents the uniform distribution over all classes, $N$ denotes the number samples in the current batch, $\alpha \in (0, 1)$ is a tunable hyperparameter controlling the emphasis between correct and incorrect predictions, and the cross-entropy function and KL divergence are defined as $\text{CE}(p, y) = -\sum_c y_c \log p_c$ and $\text{KL}(p \,\|\, q) = \sum_c p_c \log(\frac{p_c}{q_c})$, respectively. We note that a similar loss has previously been proposed in Outlier Exposure (OE) (Hendrycks et al., 2018) for out-of-distribution (OOD) sample detection. Here, the goal is to make sure that OOD examples are assigned low confidence scores by tuning the confidence on an auxiliary outlier dataset. However, to the best of our knowledge, this idea has not previously been used to improve deferral performance of a smaller model in a cascading chain.

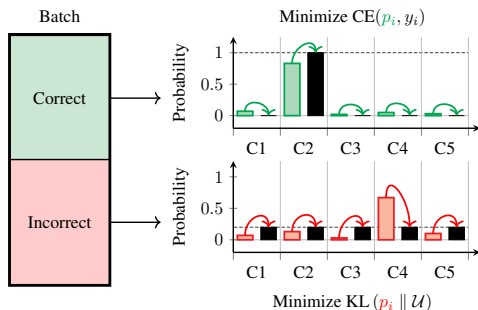

Figure 2: **GATEKEEPER Overview**: We want correctly predicted samples to maintain their current prediction by ensuring that cross entropy is decreased. At the same time, we want incorrectly predicted samples to yield a uniform confidence across classes, leading to a low overall confidence score (high predictive entropy).

We emphasize that the trade-off parameter $\alpha$ plays a critical role as part of this optimization setup as it directly influences model utility and deferral performance. A lower value of $\alpha$ emphasizes reducing confidence in incorrect predictions by pushing them closer to the uniform distribution, making the model more cautious in regions where it may make mistakes. Conversely, a higher value of $\alpha$ encourages the model to increase its confidence on correct predictions, sharpening its decision boundaries and enhancing accuracy where it is already performing well. Thus, $\alpha$ serves as a crucial hyperparameter that balances the trade-off between improving calibration by mitigating overconfidence in errors and reinforcing confidence in accurate classifications. By appropriately tuning $\alpha$, practitioners can control the model's behavior to achieve a desired balance between reliability in uncertain regions and decisiveness in confident predictions, tailored to the specific requirements of their application.

We further generalize this loss to token-based models (e.g., LMs and VLMs) where

$$\mathcal{L}_{\text{corr}} = \frac{1}{N}\sum_{\substack{i=1\\t=1}}^{N,\,T}\mathbb{1}\{y_{i,t} = \hat{y}_{i,t}\}\text{CE}(p_{i,t}(\mathbf{x}_i), y_{i,t}) \qquad \mathcal{L}_{\text{incorr}} = \frac{1}{N}\sum_{\substack{i=1\\t=1}}^{N,\,T}\mathbb{1}\{y_{i,t} \neq \hat{y}_{i,t}\}\text{KL}\left(p_{i,t}(\mathbf{x}_i) \,\|\, \mathcal{U}\right). \quad (3)$$

Here, $y_{i,t}$ and $\hat{y}_{i,t}$ denote the true and predicted tokens at position $t$ for sample $i$, $p_{i,t}$ is the predicted token distribution at position $t$ for sample $i$, and $T$ is the sequence length for the token-based model. The token-level loss ensures that correct token predictions are made confidently while incorrect tokens are assigned smaller confidences.

**Practical Computation of the Gatekeeper Loss.** We evaluate Gatekeeper *once per mini-batch* within the standard training loop—no auxiliary passes or data-set re-shuffling are required. Given a mini-batch $B = \{(x_i, y_i)\}_{i=1}^{N}$, we perform a single forward pass through $\mathcal{M}_S$. This allows us to obtain (i) a logit vector $z_i = f_\theta(\mathbf{x}_i)$, and (ii) predicted labels $\hat{y}_i = \arg\max_c z_{i,c}$. Two

binary masks $m_i^{\text{corr}} = I\{y_i = \hat{y}_i\}$ and $m_i^{\text{incorr}} = \neg m_i^{\text{corr}}$ are computed *on-the-fly*. The hybrid loss is then assembled in a fully vectorized manner with components $\mathcal{L}_{\text{corr}} = \frac{1}{N} \sum_i m_i^{\text{corr}} \text{CE}(p_i, y_i)$ and $\mathcal{L}_{\text{incorr}} = \frac{1}{N} \sum_i m_i^{\text{incorr}} \text{KL}(p_i \| \mathcal{U})$ where $p_i = \text{softmax}(z_i)$. Because both masks and losses are computed inside the same tensor graph, back-propagation incurs only the cost of $\mathcal{O}(N \times C)$ element-wise operations—identical to a vanilla cross-entropy step. This single-pass design keeps the computational overhead negligible while guaranteeing that in the full loss $\mathcal{L} = \alpha \mathcal{L}_{\text{corr}} + (1-\alpha) \mathcal{L}_{\text{incorr}}$ every sample contributes to either $\mathcal{L}_{\text{corr}}$ or $\mathcal{L}_{\text{incorr}}$ in the same optimization step.

**Stage 2 (Inference): Confidence Computation & Thresholding.** After fine-tuning $\mathcal{M}_S$ with GATEKEEPER, we apply standard confidence- and entropy-based techniques for model uncertainty to obtain a deferral signal. We use the selective prediction framework to determine whether a query point $\mathbf{x} \in \mathbb{R}^D$ should be accepted by $\mathcal{M}_S$ or routed to $\mathcal{M}_L$. Selective prediction alters the model inference stage by introducing a deferral state through a *gating mechanism* (El-Yaniv & Wiener, 2010). At its core, this mechanism relies on a deferral function $g : \mathbb{R}^D \to \mathbb{R}$ which determines if $\mathcal{M}_S$ should output a prediction for a sample $\mathbf{x}$ or defer to $\mathcal{M}_L$. Given a targeted acceptance threshold $\tau$, the resulting predictive model can be summarized as:

$$(\mathcal{M}_S, \mathcal{M}_L, g)(\mathbf{x}) = \begin{cases} \mathcal{M}_S(\mathbf{x}) & g(\mathbf{x}) \geq \tau \\ \mathcal{M}_L(\mathbf{x}) & \text{otherwise.} \end{cases} \tag{4}$$

*Classification Models (Max Softmax).* Let $\mathcal{M}_S$ produce a categorical distribution $\{p(y = c \mid \mathbf{x})\}_{c=1}^C$ over $C$ classes. Then we define the gating function as

$$g_{\text{CL}}(\mathbf{x}) = \max_{1 \leq c \leq C} p(y = c \mid \mathbf{x}). \tag{5}$$

*Token-based Models (Negative Predictive Entropy).* Let $\mathcal{M}_S$ produce a sequence of categorical distributions $\{p(y_t = c \mid \mathbf{x})\}_{c=1}^C$ for each token index $t \in T$. Then we define the gating function as

$$g_{\text{NENT}}(\mathbf{x}) = \frac{1}{T} \sum_{t=1}^T \sum_{c=1}^C p(y_t = c \mid \mathbf{x}) \log p(y_t = c \mid \mathbf{x}), \tag{6}$$

where $y_t \in [C]$ is the predicted token at time step $t$, $p(y_t = c \mid \mathbf{x})$ is the (conditional) probability of token $k$ at step $t$, and $T$ is the total number of token positions for the sequence. Across both model classes, higher values of $g_{\text{CL}}$ or $g_{\text{NENT}}$ indicate higher prediction confidence (i.e., lower uncertainty).

## 4 Experiments

In this section, we detail the experiments used to evaluate the effectiveness of GATEKEEPER across three distinct model classes: encoder-only classification models, decoder-only language models, and encoder-decoder vision-language models. Each setup involves a cascade where a smaller model can defer inputs to a larger, more capable model. Our method and all competing baselines are applied on top of the same training/fine-tuning protocol used for obtaining the initial $\mathcal{M}_S$ and $\mathcal{M}_L$ models.

### 4.1 Encoder-only Setup (Classification Models)

We comprehensively assess the performance of GATEKEEPER across different model architectures and task types, starting with image classification. We train both a large model and a small model on the following datasets: CIFAR-10/100 (Krizhevsky et al., 2009), Food-101 (Bossard et al., 2014), and TinyImageNet200 (Le & Yang, 2015). For both CIFAR datasets we use a ResNet-18 (He et al., 2016) as $\mathcal{M}_L$ and a custom CNN as $\mathcal{M}_S$. For Food-101 and TinyImageNet200 we instead use a ResNet-50 (He et al., 2016) as $\mathcal{M}_L$ and a Mobilenet V3 Small (Howard et al., 2019) as $\mathcal{M}_S$, where the latter is trained using knowledge distillation from the big model.

**Evaluation Metrics.** We measure the performance of GATEKEEPER and the resulting deferral function $g(\cdot)$ using the following performance metrics (see example in Figure 3 for an overview):

1. The **Distributional Overlap of Confidences of Correct and Incorrect Predictions** $s_o$ is defined as the integral of the minimum of the probability density functions (PDFs) of confidence scores

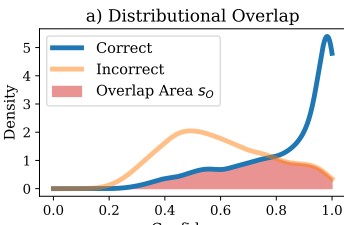 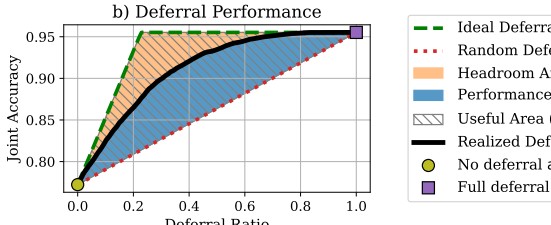

Figure 3: **Performance metrics overview**: **(a)** Distributional Overlap $s_o$: the densities of confidence scores for correctly (green) and incorrectly classified (red) samples, with the overlap area shaded in blue. Smaller values are better ($\downarrow$). **(b)** Deferral Performance $s_d$: how joint accuracy between $\mathcal{M}_S$ and $\mathcal{M}_L$ varies with deferral ratio, showing random (red), ideal (green), and realized (black) deferral strategies. The blue region shows the realized performance gain, the hatched portion represents the range of useful deferral functions, and the green region indicates the potential headroom over the realized deferral. Larger values are better ($\uparrow$).

for correctly classified samples, $\hat{p}_{\text{corr}}(c)$, and incorrectly classified samples, $\hat{p}_{\text{incorr}}(c)$ (see Figure 3a). Formally, given the confidence sets $\mathcal{C}_{\text{corr}}$ and $\mathcal{C}_{\text{incorr}}$, the overlap $s_o$ is computed as

$$s_o = \int_0^1 \min \left\{ \hat{p}_{\text{corr}}(c), \ \hat{p}_{\text{incorr}}(c) \right\} \ \mathrm{d}c, \tag{7}$$

where the PDFs are estimated using Kernel Density Estimation (KDE). If $s_o = 1$, then $\mathcal{M}_S$ cannot distinguish the confidence distribution of correct and incorrect predictions; if $s_o = 0$, then $\mathcal{M}_S$ can perfectly separate correct and incorrect predictions. Note that a related way of capturing the distributional separability is given by the Area Under the Receiver Operating Characteristic Curve (AUROC) which we discuss in Appendix C.3.

2. **Deferral Performance** $s_d$: To formally quantify how well $\mathcal{M}_S$ defers difficult inputs to $\mathcal{M}_L$, we examine the joint performance across all possible deferral ratios $r \in [0, 1]$, where $r$ denotes the fraction of inputs sent to $\mathcal{M}_L$ based on a particular threshold $\tau$ (recall Equation (4)). Figure 3 b) illustrates how, as $r$ increases from 0 to 1, the overall (joint) accuracy $\text{acc}(r)$ increases from the accuracy of $\mathcal{M}_S$ (yellow circle, no deferral) to the accuracy of $\mathcal{M}_L$ (purple square, full deferral). Useful deferral models are constrained to operate between random deferral ($\text{acc}_{\text{rand}}$, red dotted line) and ideal deferral ($\text{acc}_{\text{ideal}}$, green dashed line). The ideal deferral $\text{acc}_{\text{ideal}}$ corresponds to the oracle solution that perfectly defers examples misclassified by $\mathcal{M}_S$ and we discuss its exact functional form in Appendix B.3. We also define the realized deferral curve, $\text{acc}_{\text{real}}$, as the joint accuracy obtained under the learned deferral strategy $g(\cdot)$ employed by $\mathcal{M}_S$ and $\mathcal{M}_L$. The deferral performance metric $s_d$ is then given as:

$$s_d = \frac{A_{\text{perf}}}{A_{\text{useful}}} = \frac{\int_0^1 \left( \text{acc}_{\text{real}}(r) - \text{acc}_{\text{rand}}(r) \right) \ \mathrm{d}r}{\int_0^1 \left( \text{acc}_{\text{ideal}}(r) - \text{acc}_{\text{rand}}(r) \right) \ \mathrm{d}r}. \tag{8}$$

This ratio quantifies the fraction of the potential improvement over random deferral that has been realized by the achieved deferral strategy. Note that $s_d = 1$ indicates perfect deferral, matching the ideal strategy, while an $s_d = 0$ implies no improvement over random deferral.

3. **Accuracy of the small model** $\text{acc}(\mathcal{M}_S)$: Finally, since GATEKEEPER emphasizes patterns for distinguishing correct/incorrect examples, the model is no longer encouraged to minimize the classification loss over the full population. As a result, improving on the correct/incorrect separation task can lead to changes in utility over the full data distribution. Hence, practically useful deferral methods need to balance both deferral performance and the accuracy of $\mathcal{M}_S$.

**Results.** Our main results are shown in Figure 4. We report performance for both a baseline model (an instance of $\mathcal{M}_S$ not trained with GATEKEEPER) and small models trained with GATEKEEPER at various $\alpha$ values. We also compare against Narasimhan et al. (2022), a common cascading baseline for supervised learning tasks. For all models, we compute deferral performance and correct/incorrect separation (center, left). The strongest performance occurs at low $\alpha$s, where the model pushes outputs of incorrect examples closer to uniform. However, this comes at a cost: the small model's accuracy degrades at low $\alpha$s (right), highlighting that the model effectively "unlearns" performance on harder examples to focus on easier ones. At larger $\alpha$s, accuracy remains stable

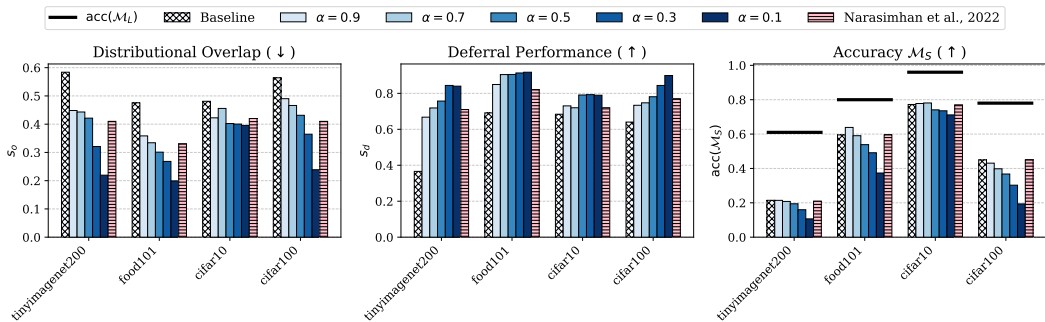

Figure 4: **Performance on image classification tasks**. We observe that lower levels of $\alpha$ lead to decreased distributional overlap between correct/incorrect predictions (left), increased deferral performance (center) and generally decreased performance over the full data distribution (right). These results support our conclusion that the small model $\mathcal{M}_S$ learns to refocus on easier subsets of the distribution while understanding more reliably when it should defer to the large model $\mathcal{M}_L$.

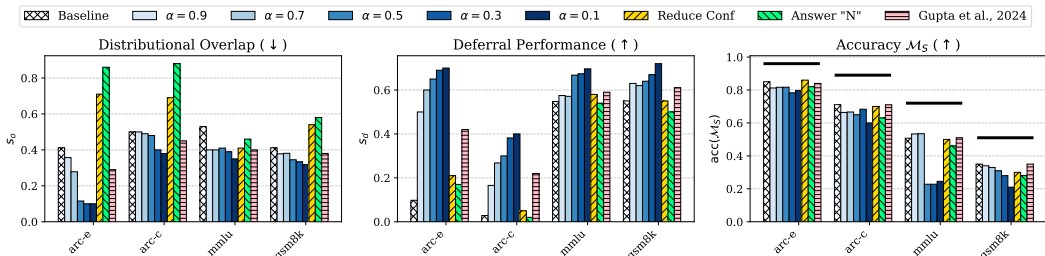

Figure 6: **Performance on language modeling tasks**. Similar as Figure 4. In addition to a non-tuned baseline, we also add an uncertainty prompting baseline, an Answer "N" option, as well as the post-hoc confidence calibration method from Gupta et al. (2024). We observe that GATEKEEPER outperforms other methods at lower levels of $\alpha$.

or improves slightly, as training emphasizes already well-predicted points. While the baseline from Narasimhan et al. (2022) preserves model accuracy, it requires explicit estimation of the expert model's correctness—necessitating either an architectural change (e.g., an added prediction head) or a separate prediction network. In contrast, GATEKEEPER relies solely on confidence tuning of the small model, making it operationally easier to deploy. This approach not only simplifies implementation but also leads to improved deferral performance by producing more reliable uncertainty estimates through a stronger correlation with correctness.

This result highlights a critical trade-off which is directly controlled by $\alpha$:

*How strongly do we want to degrade model performance over the full data distribution in order to obtain a better deferral model?*

We note that this compromise between raw model utility and deferral performance is not surprising and similar trade-offs exist in fairness (Dutta et al., 2020; Yaghini et al., 2023) and privacy (Abadi et al., 2016; Rabanser et al., 2023). We study this trade-off explicitly in Figure 5 showing (i) a clear negative correlation between deferral performance and the small model's accuracy; and (ii) a clear positive correlation between the overlap of correct/incorrect confidences and the accuracy of $\mathcal{M}_S$.

## 4.2 Decoder-Only Setup (LLMs)

In the decoder-only setup, we explore the application of LLMs. Our primary models of interest are the scalable LMs from the Gemma model class (GemmaTeam et al., 2024). We choose Gemma2B as $\mathcal{M}_S$ and Gemma7B as $\mathcal{M}_L$. Similar to the encoder-only setup, we employ smaller LMs as the initial classifiers to manage simpler next-token prediction tasks. The deferral strategy involves routing only those token sequences that exhibit high uncertainty—as determined by high predictive entropy—to the more powerful model $\mathcal{M}_L$.

Our experiments begin by taking the instruction-tuned checkpoints of Gemma2B and Gemma7B and fine-tuning both models on the training split of each dataset to ensure that the model (i) performs well on the task and (ii) is familiar with the desired response format. This step is performed using standard supervised fine-tuning. Next, we fine-tune $\mathcal{M}_S$ with GATEKEEPER on the same training split to reduce confidence on incorrect next-token predictions. Finally, we evaluate the model trained with GATE-KEEPER on a validation split. The datasets used are ARC-e/c (Clark et al., 2018), MMLU (Hendrycks et al., 2020), and GSM8K (Cobbe et al., 2021). The evaluation metrics for our LLM experiments match those used in Section 4.1.

**Results.** We present our main results in Figure 6, comparing the baseline model's deferral and correct/incorrect separation ability to our fine-tuned model across different $\alpha$s. We observe a similar trend as in the image classification setting: higher $\alpha$s maintain raw prediction performance closer to the baseline but offer limited gains in separation, while lower $\alpha$s improve deferral more substantially at the cost of overall accuracy. In addition to the baseline model (not fine-tuned with GATEKEEPER), we include results from Gupta et al. (2024) (an extension of Narasimhan et al. (2022) to token-based sequence models), as well as two uncertainty prompting baselines (described in Appendix C.2): (i) *Reduce Confidence*, which appends instructions to encourage the model to lower confidence when uncertain; and (ii) *Answer "N"*, which instructs the model to respond with "N" if uncertain. We find that GATEKEEPER outperforms Gupta et al. (2024) in terms of correct/incorrect sepration and deferral at the cost of overall utility. Consistent with prior findings from Kadavath et al. (2022), the prompting baselines do not reliably improve deferral.

## 4.3 Encoder-Decoder Setup (VLMs)

Finally, we examine models with both visual and textual processing capabilities, ideal for tasks requiring joint image understanding and language generation. We use the PaliGemma (Steiner et al., 2024) model family—encoder-decoder models designed for VL tasks such as image captioning, visual question answering, and descriptive image classification. The encoder processes input images into rich feature representations, while the decoder generates textual outputs. We use PaliGemma1B as $\mathcal{M}_S$ and PaliGemma7B as $\mathcal{M}_L$. Our deferral strategy runs the smaller VLM on all inputs and only defers to the more bigger 7B model when $\mathcal{M}_S$'s predictive entropy falls below a set threshold.

Similar to our experiments on LMs in Section 4.2, we employ two stages of fine-tuning. First, we take the instruction-tuned checkpoints of PaliGemma1B and PaliGemma7B and then fine-tune both models on the training split of a given dataset. Next, we fine-tune only $\mathcal{M}_S$ using GATEKEEPER before evaluating the model on a validation/test split of the dataset. The datasets we consider are two classification datasets (VQAv2 (Goyal et al., 2017), AI2D (Hiippala et al., 2021)) and two captioning datasets (Cococap (Lin et al., 2014), Screen2Words (Wang et al., 2021)). This allows us to evaluate GATEKEEPER in closed-form vision-language classification setups and open-form text generation.

**Factuality Scoring.** For classification tasks we apply our analysis in the same way as in Section 4.2. However, for captioning datasets we need to evaluate the quality of a caption generated by PaliGemma. To do that, we compute a factuality score which judges whether the generated caption is semantically coherent with respect to a reference caption using the Gemini LLM (GeminiTeam et al., 2023).

Figure 5: **Performance trade-off between small model accuracy** $\mathbf{acc}(\mathcal{M}_S)$ **and deferral evaluation metrics**. The baseline model obtained without fine-tuning using GATE-KEEPER is often the most accurate model over the full data distribution. With the introduction of GATEKEEPER we can improve distinguishability of correct/incorrect predictions (top) as well as deferral performance (bottom) at the expense of model utility. Successful cascading solutions in practice need to balance both maintaining high model accuracy and deferral performance.

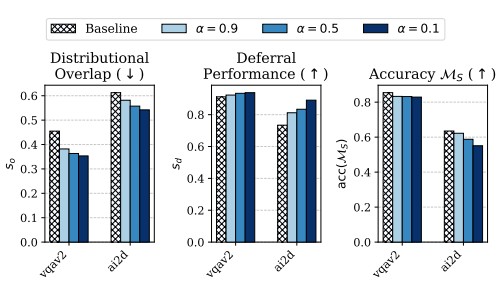
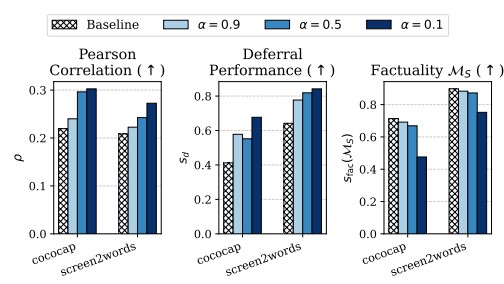

Figure 7: **Performance on VLM classification (left) and captioning tasks (right)**. Consistent with results in Figures 4 and 6, we see that smaller $\alpha$s lead to improved deferral performance.

Specifically, the Gemini LLM is prompted with an instruction of the form: *"Are these captions semantically equivalent?"*, followed by both the candidate caption and the reference caption. The model then responds with either *"Yes"* or *"No"*. Finally, we compute the log-likelihood of each response and normalize it to a probability, reflecting the LLM's confidence in the captions being factually aligned. We detail this process in Appendix C.4 and denote the factuality score for input point $\mathbf{x}_i$ with candidate caption $\hat{\mathbf{y}}_i$ and ground truth caption $\mathbf{y}_i$ as $s_{\text{Fac}}(\hat{\mathbf{y}}_i, \mathbf{y}_i)$.

**Measuring Correlation Between Factuality and Negative Predictive Entropy.** Since the result of evaluating $s_{\text{Fac}}(\hat{\mathbf{y}}_i, \mathbf{y}_i)$ is no longer binary, our evaluation metrics which previously relied on accuracy cannot be used directly to evaluate deferral performance and the correct/incorrect entropy distribution separation. We address this issue by replacing the distributional overlap computation with the Pearson correlation $\rho(g_{\text{NENT}}(\mathbf{x}_i), s_{\text{Fac}}(\hat{\mathbf{y}}_i, \mathbf{y}_i))$ between the negative predictive entropy of a caption $g_{\text{NENT}}(\mathbf{x}_i)$ and its associated factuality score $s_{\text{Fac}}(\hat{\mathbf{y}}_i, \mathbf{y}_i)$. We also adapt our deferral performance metric from Equation (8) to rely on factuality measures instead of accuracy.

**Results.** We present our results in Figure 7, comparing the baseline model's deferral ability to our fine-tuned models across different $\alpha$s. For classification (Figure 7, left), we observe the same trends as in previous classification and language modeling experiments. For captioning (Figure 7, right), GATEKEEPER increases the correlation between factuality and negative predictive entropy, enabling better deferral from $\mathcal{M}_S$ to $\mathcal{M}_L$ as $\alpha$ decreases. This shows that our method generalizes beyond closed-form classification to open-form sequence generation tasks. While we also benchmarked the prompting baselines from Section 4.2, PaliGemma did not return responses for the modified prompts—likely due to its rigid pretraining and prompting instructions (Beyer et al., 2024).

## 5 Conclusion

In this work we present a novel loss function called GATEKEEPER for improving confidence calibration in a cascade between a small local and a larger remote model. Our loss is architecture and task agnostic, making it flexibly applicable across a wide range of applications. Our results on encoder-only classification models, decoder-only language models, and encoder-decoder vision-language models demonstrate that our approach improves over standard confidence-based deferral rules and effectively leads the small model to unlearn how to handle complex queries in favor of easier ones.

**Limitations.** While our approach demonstrates promising results, several limitations remain. First, we assume that only the smaller model can be fine-tuned, whereas in some applications the larger model could also be adjusted to improve deferral. Second, in language settings, GATEKEEPER may be overly aggressive: multiple token sequences can convey the same meaning, so penalizing deviations based on exact tokens rather than semantics may be suboptimal. Ideally, deferral should account for semantic correctness rather than surface-level mismatches. Third, we did not extensively evaluate across diverse model families in LLM and VLM settings, though we did include such comparisons for classification tasks. Fourth, our evaluation focusses on a two-stage cascade consisting a small model and a big model. However, we are confident that GATEKEEPER can scale to multi-stage setups (see Appendix B.5). Finally, our use of a generative model (e.g., Gemini) to evaluate captioning introduces the possibility of erroneous judgments, as LLMs are themselves imperfect evaluators.

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

# A    Broader Impact

This work contributes to the responsible and efficient deployment of machine learning systems by improving the decision-making capabilities of smaller, local models in model cascade architectures. By introducing a loss function that calibrates model confidence with respect to correctness, our approach enhances both the performance and transparency of automated systems that must decide when to act autonomously and when to defer to a more capable model. This design can improve the accessibility and sustainability of machine learning applications by reducing reliance on large, energy-intensive models—particularly important in low-resource environments or edge computing.

At the same time, the ability to fine-tune smaller models to strategically abstain from uncertain predictions raises important considerations for fairness and accountability. In high-stakes applications such as healthcare or finance, improper tuning of the deferral threshold—or uncalibrated confidence estimates—could lead to the systematic denial of service or misallocation of computational resources. Care must be taken to ensure that such systems are thoroughly evaluated not only for average performance but also for differential performance across subgroups. Moreover, the use of large models as fallback decision-makers assumes their correctness, which may not always hold, especially in underrepresented domains. We therefore encourage developers and practitioners to accompany deployments of cascade-based systems with rigorous audits of fairness, reliability, and alignment with human values.

# B    Additional Background

## B.1    Related Work

### B.1.1    LLM Routing

Ding et al. (2024) propose a hybrid LLM inference pipeline that routes each query either to a small on-device model or a larger high-quality model based on the query's predicted difficulty and a tunable quality threshold. This cost-aware router allows dynamically trading off accuracy for efficiency, enabling up to a 40% reduction in expensive model calls without degrading answer quality. Similarly, Shnitzer et al. (2023) present a method to select the best model from a pool of pre-trained LLMs for each input by learning a "router" on many benchmark tasks. Without requiring labeled examples from the new target task, their approach uses existing datasets to train input-based model selectors, which consistently outperform always using the single best LLM for all queries.

### B.1.2    Model Cascade Learning

Nie et al. (2024) introduce an online cascade-learning framework where lightweight models are incrementally trained to imitate a powerful LLM's decisions on a data stream, deferring to the LLM only when necessary. They cast cascade construction as an imitation-learning problem with theoretical no-regret guarantees, achieving LLM-level accuracy while cutting inference cost by up to 90% and maintaining robustness to distribution shifts over time. Chen et al. (2023b) outline strategies for reducing LLM usage cost and present *FrugalGPT*, a cascade approach that learns to route queries through combinations of smaller or larger LLMs to balance cost and performance. Their experiments show that an adaptive use of multiple models can match the accuracy of the strongest individual LLM (e.g., GPT-4) with up to 98% cost savings. It can also slightly exceed GPT-4's accuracy at equal cost, highlighting the benefit of cascades that allocate queries to the most appropriate model for each input.

### B.1.3    Confidence Calibration in LLMs

Jitkrittum et al. (2023) analyze the classical strategy of confidence-based deferral in model cascades, wherein a model hands off to a stronger model if its confidence is below a threshold, to determine when this simple strategy succeeds or breaks down. They derive the optimal deferral policy in theory and show that naïve confidence thresholds perform well in general but can fail when later models are specialists (only reliable on certain inputs), when there is label noise, or under distribution shift—scenarios where more sophisticated deferral criteria yield better performance. Geng et al. (2023) provide a comprehensive survey of methods for confidence estimation and calibration in LLM outputs. They review recent techniques to quantify uncertainty in large language model predictions, discuss challenges unique to LLMs, and highlight advancements that improve alignment between

a model's reported confidence and its actual accuracy across tasks. Azaria & Mitchell (2023) find evidence that an LLM's internal activations encode whether or not it is producing a truthful answer, even when the model's output is incorrect or fabricated. By training a classifier on the model's hidden state (without fine-tuning the LLM itself), they can often detect when the model is "lying" or unsure, suggesting that large models internally recognize their mistakes or uncertainty despite outwardly confident responses. Similarly, Liu et al. (2024) propose a supervised approach to LLM uncertainty quantification that leverages labeled examples and the model's hidden representations to predict the correctness of its answers. They show that incorporating features from the model's internal layers yields significantly improved uncertainty estimates and calibration across diverse tasks, with these gains transferring robustly to new domains. Notably, their method is easy to implement and can be adapted to different levels of model access (black-box vs. white-box), making it widely applicable.

### B.1.4 Confidence Verbalization in LLMs

Lin et al. (2022) demonstrate that GPT-3 can be fine-tuned to output a calibrated verbal confidence (e.g., "I'm 90% sure") along with each answer. This model's stated confidence levels align well with its true correctness likelihood and remain fairly well-calibrated even under distribution shift, marking the first instance of an LLM explicitly expressing useful uncertainty estimates in natural language. Xiong et al. (2024) thoroughly evaluate black-box methods for eliciting an LLM's self-reported confidence through prompting and answer sampling. They find that current LLMs tend to verbalize overly high confidence (mirroring human overconfidence), but that carefully designed prompts, consistency checks across multiple sampled answers, and improved aggregation strategies can mitigate this issue. Moreover, larger models generally show better calibration and an improved ability to predict their own failures, though room for further improvement remains in making their expressed uncertainty truly reliable. Mielke et al. (2022) examine whether a conversational agent's expressed certainty corresponds to its actual knowledge, showing that off-the-shelf dialogue models are poorly "linguistically calibrated." They demonstrate that a model's likelihood of giving a correct answer can be estimated via an auxiliary model and used as a control signal to adjust the agent's responses. The resulting dialogue agent exhibits far less overconfident language when it is likely to be wrong, improving transparency about uncertainty in its answers. Finally, Mahaut et al. (2024) assess the reliability of various methods to estimate an LLM's *factual confidence* – the probability that its answer is correct – under both in-domain and paraphrased inputs. Through a rigorous evaluation on QA and fact-checking tasks, they conclude that the most trustworthy confidence scores come from model-introspective approaches (e.g., a trained probe on hidden states), albeit at the cost of requiring full model access and training data. They also highlight that an LLM's confidence can be unstable under meaning-preserving input variations (paraphrases), underscoring the need for more robust and stable confidence estimation techniques for factual correctness.

### B.1.5 Speculative Decoding

As noted in the introduction and our related work section, there exists a connection between model cascading and speculative decoding. Speculative decoding is a technique to accelerate inference by pairing a small, fast "draft" model with a larger target model. The draft model generates multiple candidate tokens in parallel, and the large model then verifies them in a single forward pass by accepting valid tokens and rejecting others. This reduces the number of expensive calls to the large model, often yielding significant speedups without sacrificing output quality. We briefly describe how this line of work relates to our goal of deferral confidence tuning.

Speculative decoding accelerates every input by asking the large model to verify draft tokens that the small model proposes. As a result, the large model is still invoked on all inputs. Our cascade, by contrast, aims to avoid calling the large model on inputs the small model can already solve. Consequently, speculative decoding optimizes in-place token-level latency, while our method optimizes end-to-end compute and monetary cost at the request level. Because the two techniques improve fundamentally different bottlenecks we do not directly compare against this class of methods. However, we note that both cascading and speculative decoding can be composed: after our deferral gate decides to consult the large model, one could still decode that portion of the input speculatively. We direct the interested reader to Narasimhan et al. (2025) for an example of such hybrids.

### B.1.6 Early Exiting

Early exit networks are models augmented with intermediate classifiers that allow predictions to be made before reaching the final layer. At inference time, the model can stop early on "easy" inputs while continuing deeper for "harder" ones, reducing computation without compromising much accuracy. This adaptive approach makes them well-suited for resource-constrained or latency-sensitive applications. As we discuss below, this class of approaches is, while related, ultimately still distinct from our goal for the following reasons:

1. Early–exiting aims to skip the remaining layers of a single network once an intermediate classifier is sufficiently confident, thereby offering a layer-level latency-accuracy trade-off within one model. Our work tackles the fundamentally different problem of model-level deferral between a small and a large model. The goal is to keep the large model entirely idle for easy requests and to invoke it only when needed. Hence our primary measure of success is joint accuracy versus cross-model compute budget, not marginal delay per layer inside a fixed backbone.

2. Early–exit usually presupposes white-box control over the entire network architecture so that branch classifiers can be inserted and jointly trained (e.g., BranchyNet (Teerapittayanon et al., 2016), Adaptive Neural Networks (Bolukbasi et al., 2017b)). Our cascade setting explicitly targets heterogeneous or API-based experts—large language models, vision–language models, and so forth—that cannot be modified. Early-exits cannot be applied in such black-box situations, whereas our GATEKEEPER fine-tuning remains feasible. Moreover, early-exiting has seen limited success for autoregressive sequence generation, where every token depends on the full hidden state; in contrast, our experiments span both classification and open-ended generation tasks.

3. Because early-exit and cascading operate at different granularities, they are complementary rather than competing techniques. One could in principle insert early exits inside the small model and still rely on calibrated deferral to the large model—yielding a three-level system (exit-1 → exit-2 → large model). Evaluating that combined design is a compelling direction for future engineering work, but it lies beyond the scope of our current study, whose contribution is a confidence-tuning loss independent of architectural changes.

### B.2 Model Access Levels

In Figure 8, we show a schematic overview of different model access levels discussed in Section 2.

### B.3 Ideal Deferral Curve

We present the functional form of the *ideal deferral* curve, denoted $\mathrm{acc}_{\mathrm{ideal}}(r)$, for a small (student) model $\mathcal{M}_S$ and a large (teacher) model $\mathcal{M}_L$. Recall that $r \in [0, 1]$ denotes the deferral ratio, i.e., the fraction of inputs that $\mathcal{M}_S$ "defers" to $\mathcal{M}_L$. Let $p_s = \mathrm{acc}(\mathcal{M}_S)$, and $p_l = \mathrm{acc}(\mathcal{M}_L)$ with $0 \leq p_s \leq p_l \leq 1$. Our goal is to describe the maximum achievable joint accuracy if exactly a fraction $r$ of the data is deferred to the large model.

**Intuition and Setup** Since $\mathcal{M}_S$ achieves accuracy $p_s$, it misclassifies a fraction $(1 - p_s)$ of the inputs. In an *ideal* scenario, we defer exactly those inputs that $\mathcal{M}_S$ is going to misclassify. Because $\mathcal{M}_L$ is more accurate ($p_l \geq p_s$) every example misclassified by $\mathcal{M}_S$ benefits from being passed to $\mathcal{M}_L$.

- **Case 1:** $r \leq (1 - p_s)$.
  We can use our entire deferral "budget" $r$ to cover only those inputs $\mathcal{M}_S$ would get wrong. Hence, deferring a fraction $r$ of the data (all from $\mathcal{M}_S$'s mistakes) raises the overall accuracy by substituting $\mathcal{M}_S$'s errors with $\mathcal{M}_L$'s accuracy $p_l$ on that fraction.

- **Case 2:** $r > (1 - p_s)$.
  We have enough capacity to defer *all* of $\mathcal{M}_S$'s mistakes, so the joint accuracy saturates at $p_l$. Deferring *additional* examples (which $\mathcal{M}_S$ would have classified correctly) will not improve the overall accuracy beyond $p_l$.

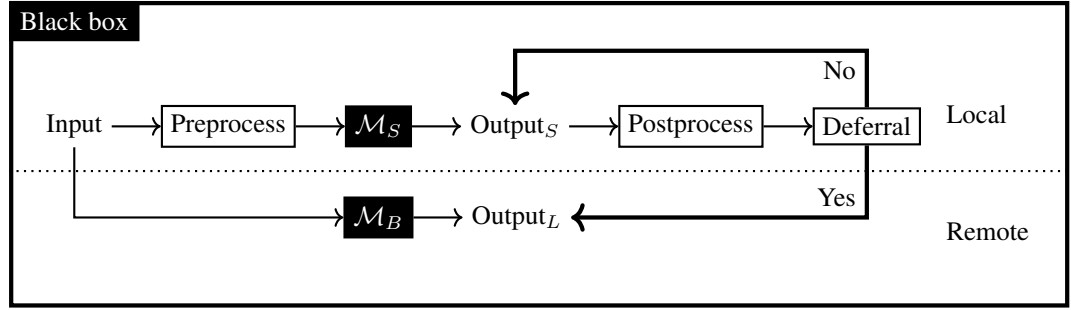

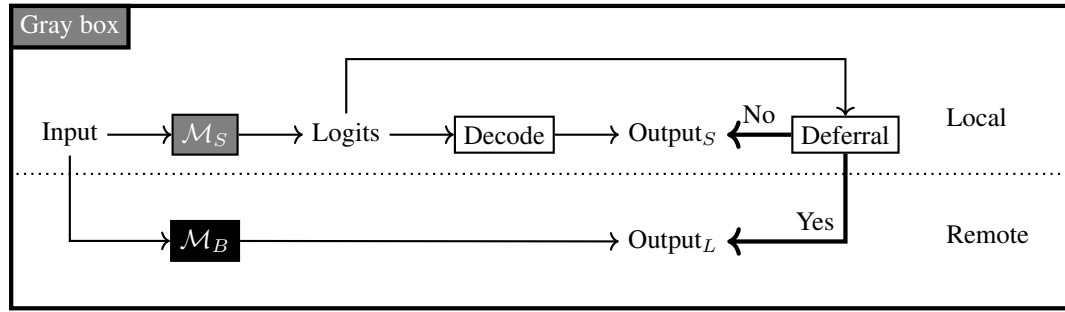

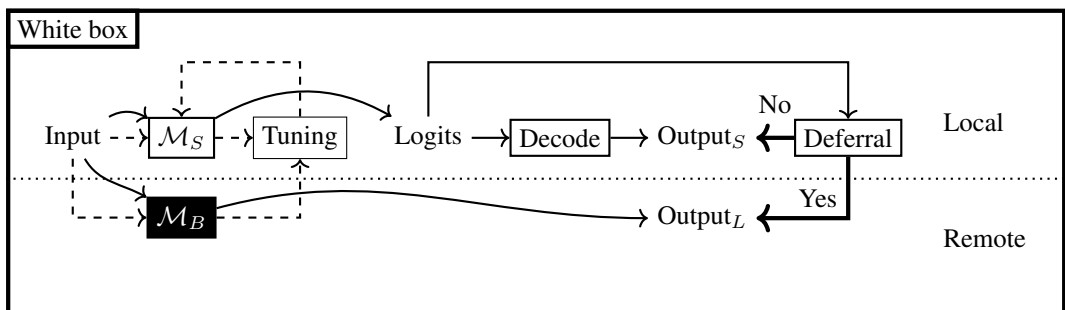

Figure 8: **An overview of different uncertainty quantification strategies depending on model access level**.

**Piecewise Functional Form**  Thus, the *ideal deferral* curve can be expressed as:

$$\text{acc}_{\text{ideal}}(r) = \begin{cases} p_s + \dfrac{p_l - p_s}{1 - p_s} \, r, & 0 \le r \le (1 - p_s), \\ p_l, & (1 - p_s) < r \le 1. \end{cases} \tag{9}$$

When $0 \le r \le (1 - p_s)$, the overall accuracy grows linearly from $\text{acc}_{\text{ideal}}(0) = p_s$ to $\text{acc}_{\text{ideal}}(1 - p_s) = p_l$. Past $r = (1 - p_s)$, it remains constant at $p_l$.

Figure 3 (b) in the main paper plots this ideal deferral curve (green line). It serves as an upper bound on how effective any real deferral strategy can be. In contrast, a purely random deferral strategy produces a linear interpolation (the red line), which is strictly below the ideal curve for most $r$. Consequently, the difference $\text{acc}_{\text{ideal}}(r) - \text{acc}_{\text{rand}}(r)$ represents the *maximum possible* gain one can achieve by carefully selecting which examples to defer rather than choosing them at random.

**Summary**  We summarize the key take-aways below:

- **Ideal Deferral Routes All Mistakes:** Only the inputs misclassified by $\mathcal{M}_S$ get deferred, guaranteeing the highest possible joint accuracy at each deferral level $r$.

- **Piecewise Definition:** Accuracy increases linearly from $p_s$ to $p_l$ over the interval $r \in [0, (1 - p_s)]$, then remains at $p_l$.

- **Upper Bound on Realized Deferral:** No actual strategy can exceed this ideal curve, as it assumes perfect knowledge of which specific inputs $\mathcal{M}_S$ would misclassify.

### B.4 Gatekeeper in the Context of Canonical Calibration Objectives

**Motivation.** Section 4 showed that the GATEKEEPER loss improves *deferral performance* with minimal implementation effort. Because many calibration objectives also manipulate confidence, we now position GATEKEEPER relative to four widely–used losses.

**Canonical calibration objectives.** Let $p_\theta(y \mid \mathbf{x})$ denote the softmax (or token) distribution predicted by a model with parameters $\theta$ and let $y^\star$ be the ground-truth label. Below we recap four popular alternate calibration objectives.

(a) **Temperature scaling** (Guo et al., 2017) applies a single scalar $T>0$ at test time: $p_T(y \mid \mathbf{x}) \propto \exp(z_c/T)$. It preserves the rank ordering and therefore *cannot tighten* the ranking-based risk–coverage curve, but can improve threshold-based acceptance.

(b) **Focal loss** (Lin et al., 2017) adds a down-weighting factor to easy examples, $\mathrm{FL}(p, y^\star) = -(1 - p_{y^\star})^\gamma \log p_{y^\star}$ with $\gamma \in [0, \infty)$. It improves class imbalance calibration but does not explicitly penalize over-confidence on *incorrect* samples.

(c) **Confidence penalty** (Pereyra et al., 2017) regularises high-entropy predictions through $\mathrm{CE}(p, y^\star) + \lambda \mathcal{H}(p)$. While it flattens *all* distributions, it does not distinguish between correct and incorrect cases.

(d) **Outlier exposure (OE)** (Hendrycks et al., 2018) adds a KL-uniform loss on auxiliary OOD data, mirroring the second term of GATEKEEPER but only on outliers, not in-distribution misclassifications.

**How Gatekeeper differs.** GATEKEEPER combines *two complementary gradients*: (i) a standard CE term on correct predictions *with an instance-level mask*, thereby **sharpening** those logits; (ii) a KL-to-uniform term on *incorrect* predictions, **flattening** their confidence. This asymmetric design forces the scalar summary $g(\mathbf{x}) = \max_c p_\theta(y{=}c \mid \mathbf{x})$ (or token-entropy) to separate correct from incorrect points *without* requiring additional heads, auxiliary datasets, or test-time tuning.

### B.5 Extension to Multi-Level Cascades

Cascading can extend beyond a single deferral $\mathcal{M}_S \to \mathcal{M}_L$ to multiple deferrals $\mathcal{M}_1 \to \mathcal{M}_2 \to \cdots \to \mathcal{M}_K$. GATEKEEPER makes no explicit assumption on the number of cascade levels—it is inherently modular and agnostic to model architecture. In practice, one can fine-tune each model $\mathcal{M}_i$ in the chain independently using the GATEKEEPER loss with a stage-specific hyper-parameter $\alpha_i$ that governs the balance between confident acceptance and deferral. A natural hierarchical training procedure is to first apply GATEKEEPER to $\mathcal{M}_1$ to calibrate its confidence, then apply it to $\mathcal{M}_2$ on the subset of inputs deferred by $\mathcal{M}_1$, and so on through $\mathcal{M}_K$. This approach scales linearly in the number of levels and incurs only the familiar overhead of per-model fine-tuning—no joint optimization or additional routing networks are required. We therefore expect that deeper cascades will offer flexible trade-offs between computational cost and predictive accuracy across multiple model tiers, and we leave a full empirical evaluation of $K > 2$ cascades to future work.

### B.6 Extension to Multi-Level Cascades

## C Additional Experimental Details

### C.1 CNN Used in Image Classification Experiments

Below we include a representation of the `SmallCNN` model used as $\mathcal{M}_S$ in image classification experiments discussed in Section 4.1:

```
SmallCNN(
  (features): Sequential(
    (0): Conv2d(3, 16, kernel_size=(3, 3), stride=(1, 1), padding=(1,
      1))
```

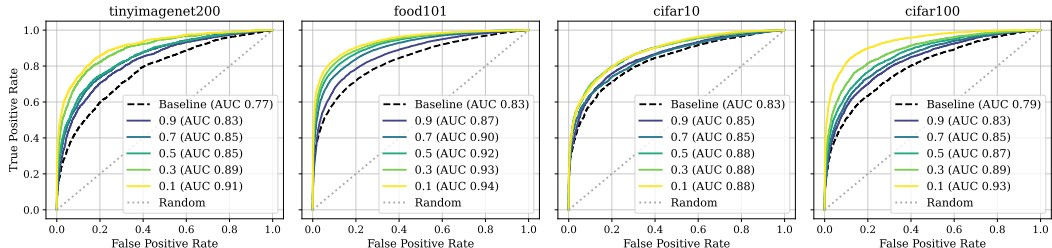

Figure 9: **ROC curves for image classification experiments**. Each figure shows the ROC curves for each of the datasets considered in Section 4.1. We observe that GATEKEEPER consistently increases separation of correct and incorrect confidence scores across varying $\alpha$ (colored curves) compared to the baseline (denoted with black dashed line).

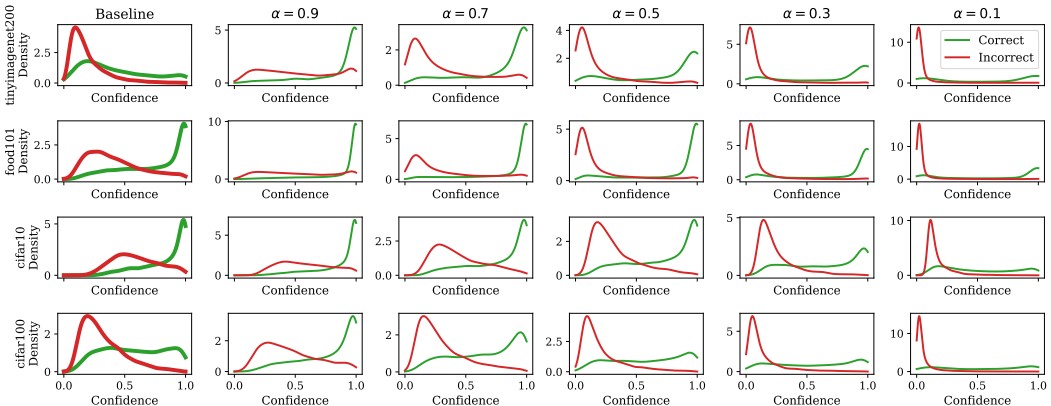

Figure 10: **Distributional overlap for image classification experiments**. Left-most column shows the results obtained using the untuned baseline, while the remaining columns correspond to the results obtained using GATEKEEPER with decreasing $\alpha$ values. Rows correspond to the datasets considered in Section 4.1. We see that GATEKEEPER increases separation of correct and incorrect confidence scores compared to the baseline.

```
4       (1): BatchNorm2d(16, eps=1e-05, momentum=0.1, affine=True,
            track_running_stats=True)
5       (2): ReLU(inplace=True)
6       (3): MaxPool2d(kernel_size=2, stride=2, padding=0, dilation=1,
            ceil_mode=False)
7       (4): Conv2d(16, 32, kernel_size=(3, 3), stride=(1, 1), padding=(1,
            1))
8       (5): BatchNorm2d(32, eps=1e-05, momentum=0.1, affine=True,
            track_running_stats=True)
9       (6): ReLU(inplace=True)
10      (7): MaxPool2d(kernel_size=2, stride=2, padding=0, dilation=1,
            ceil_mode=False)
11    )
12    (classifier): Sequential(
13      (0): Linear(in_features=2048, out_features=64, bias=True)
14      (1): ReLU(inplace=True)
15      (2): Linear(in_features=64, out_features=10, bias=True)
16    )
17  )
```

## C.2 Reduce Confidence and Answer "N" Baselines

In addition to the baseline model in Section 4.2 (i.e., a model that was not fine-tuned with our specialized $\mathcal{L}_{\text{def}}$ loss but from which we still compute predictive entropy as a deferral signal), we also examine two additional methods aimed at eliciting uncertainty from the model directly via prompt

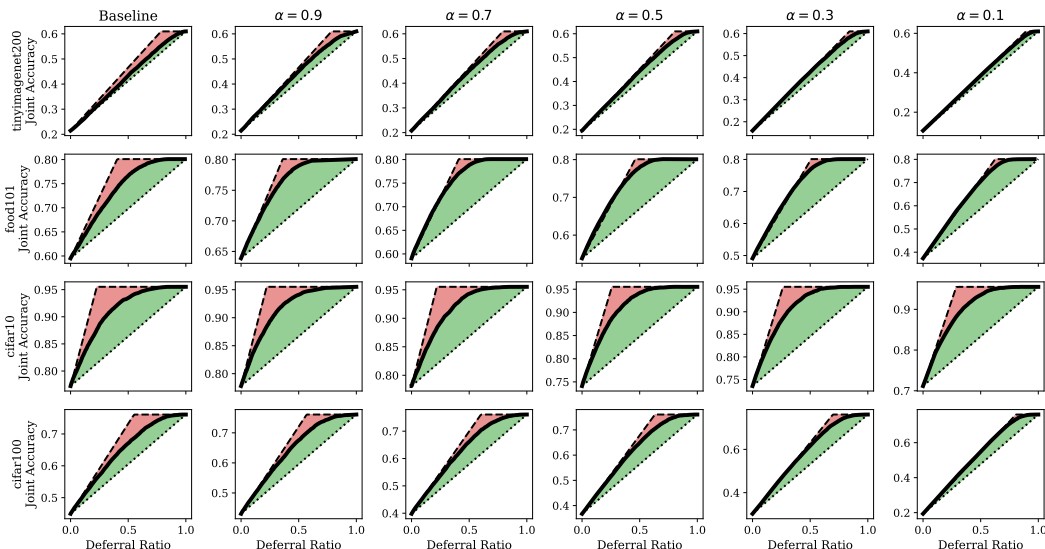

Figure 11: **Deferral curves for image classification experiments**. Left-most column shows the results obtained using the untuned baseline, while the remaining columns correspond to the results obtained using GATEKEEPER with decreasing $\alpha$ values. Rows correspond to the datasets considered in Section 4.1 The results show that GATEKEEPER brings the realized deferral (black line) closer to the ideal deferral (dashed upper line).

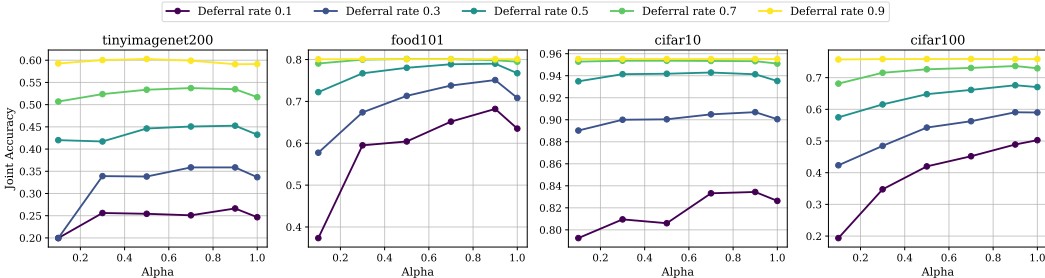

Figure 12: **Joint accuracy across different levels of** $\alpha$. For varying fixed deferral ratios, we observe that the accuracy of $\mathcal{M}_S$ generally decreases as $\alpha \to 0$.

modifications. Both methods are *black box* approaches that only rely on a query interface to the model via prompt injection, and we provide their implementation details below.

**Reduce Confidence.** In this setting, we modify the original prompt $\mathbf{x}$ by appending an additional instruction $\mathbf{x}'$ that encourages the model to respond with lower confidence when it is uncertain: $\mathbf{x} \leftarrow \mathbf{x} \mid \mathbf{x}'$. For instance, the instruction we add is:

$$\mathbf{x}' = \text{``Respond with low confidence if you are uncertain.''}$$

We treat this appended text as a hint to the model to self-regulate its confidence when producing an answer. This is similar in spirit to other black box approaches such as confidence quantification, rejection awareness, remote model notice, and self-critiquing. Although Xiong et al. (2024) show that large language models can express aspects of their confidence via prompting, our experiments indicate that simply prompting the model to express lower confidence does not reliably improve the separation of correct versus incorrect predictions, nor does it offer advantages in a deferral setting. These findings are in line with those reported in Kadavath et al. (2022).

**Answer "N."** We also consider an alternate prompt modification, in which the appended instruction is:

$$\mathbf{x}' = \text{``Respond with `N' if you are uncertain.''}$$

This approach explicitly instructs the model to produce a special "N" token to indicate uncertainty or lack of confidence. The intuition is that by introducing a designated "uncertain" response, one might isolate uncertain cases for deferral. However, our results in Section 4.2 similarly show that the model's ability to follow this instruction is inconsistent and does not substantially improve performance as a deferral model. The model often remains overconfident and fails to produce "N" in cases where it is in fact incorrect.

## C.3 Additional metrics

In addition to the metrics outlined in Section 4, we also consider the **Area Under the Receiver Operating Characteristic Curve** (AUROC) ($s_{\text{AUROC}}$). The AUROC quantifies the model's ability to discriminate between correctly and incorrectly classified data points by evaluating the trade-off between the True Positive Rate (TPR) and the False Positive Rate (FPR) across various confidence thresholds $\tau$. Formally, given the confidence sets $\mathcal{C}_{\text{corr}}$ and $\mathcal{C}_{\text{incorr}}$, the AUROC is defined as

$$s_{\text{AUROC}} = \int_0^1 \text{TPR}(\tau) \, d\text{FPR}(\tau), \tag{10}$$

where for each threshold $\tau \in [0,1]$ we compute $\text{TPR}(\tau) = \frac{|\{c \in \mathcal{C}_{\text{corr}} \mid c \geq \tau\}|}{|\mathcal{C}_{\text{corr}}|}$ and $\text{FPR}(\tau) = \frac{|\{c \in \mathcal{C}_{\text{incorr}} \mid c \geq \tau\}|}{|\mathcal{C}_{\text{incorr}}|}$. Note that $s_{\text{AUROC}} = 1$ indicates perfect separability and $s_{\text{AUROC}} = 0.5$ corresponds to a random guessing baseline.

## C.4 Factuality Scoring

Factuality scoring with Gemini for a reference caption $r$ and a candidate caption $c$ is computed as follows:

1. **Compute the log-likelihoods.** Let $\ell_{\text{Same}}(c, r)$ be the log-likelihood that the model outputs "Same" for a given candidate caption $c$ and reference $r$, and let $\ell_{\text{Diff}}(c, r)$ be the log-likelihood that the model outputs "Different".

2. **Apply softmax.** To convert these log-likelihoods into probabilities, we exponentiate and normalize:
$$p(\text{Same} \mid c, r) = \frac{\exp\big(\ell_{\text{Same}}(c, r)\big)}{\exp\big(\ell_{\text{Same}}(c, r)\big) + \exp\big(\ell_{\text{Diff}}(c, r)\big)},$$
$$p(\text{Diff} \mid c, r) = \frac{\exp\big(\ell_{\text{Diff}}(c, r)\big)}{\exp\big(\ell_{\text{Same}}(c, r)\big) + \exp\big(\ell_{\text{Diff}}(c, r)\big)}.$$

3. **Interpret the probability.** The value $p(\text{Same} \mid c, r)$ is then taken as the factual alignment score, expressing how confidently the model believes the candidate caption is factually aligned with the reference.

## C.5 Additional Experimental Results

In this section, we provide additional experimental results further supporting our findings reported for image classification experiments in Section 4.1. In particular, we show ROC curves in Figure 9 and distributional overlap in Figure 10, both demonstrating that GATEKEEPER increases the separation of correct/incorrect confidence scores. Similarly, the deferral curves in Figure 11 clearly show that GATEKEEPER successfully pushed the realized deferral (black line) closer to the ideal one (marked with dashed upper line). Lastly, we report the joint accuracy of $\mathcal{M}_S$ across varying $\alpha$ parameter in Figure 12. As discussed in Section 4, we observe that $\mathcal{M}_S$'s accuracy generally decreases with $\alpha \to 0$.

