# OpenReview forum: "Gatekeeper: Improving Model Cascades Through Confidence Tuning"
_NeurIPS.cc/2025/Conference — NeurIPS 2025 poster_

### Official Review · Reviewer_aLX9 · 2025-06-15

**Clarity:** 3
**Significance:** 2
**Originality:** 2
**Rating:** 4
**Confidence:** 4

**Summary:**

This paper aims to solve the problem in model cascades, which often deploy small models along with large models and rely on routing and deferral mechanisms to offload requests to small models and only escalate the requests to large models if the small models are not confident enough. This paper tries to tame the tradeoff between model accuracy and deferral accuracy across many model architectures and workloads. The core of this paper is a general-purpose loss function, Gatekeeper, which helps to fine-tune the small model to calibrate the small model’s confidence to improve the efficacy of model cascades.

**Questions:**

Please also refer to my comments above for my full review.
1. Is it correct that the tuning of $\alpha$ and switching of the large model will require a new round of fine-tuning?
2. Does your new loss function break the performance of small models, especially for decoder-only models?
3. How does this paper compare with related works
    3.1 Compare with speculative decoding in terms of latency savings for generative models.
    3.2 Compare with other methods (e.g., early exit) in terms of latency savings for classification workloads.

**Ethical Concerns:**

["NO or VERY MINOR ethics concerns only"]

**Final Justification:**

Most of my conceptual questions have been resolved by the authors. I only have minor issues with the finetune related overheads and how to handle the inputs that can't be solved by both small and large models.

**Limitations:**

Yes

**Paper Formatting Concerns:**

No concerns regarding formatting.

**Quality:**

2

**Strengths And Weaknesses:**

**Strengths**
1. The paper is well written and easy to follow.
2. The evaluation covers diverse model architectures (encoder only, decoder only, and encoder-decoder models) and tasks (classification and generation).

**Weaknesses**
1. As the paper indicates, the core contribution of this paper, the loss function, which has been studied and applied in previous works. I don’t think there are unique challenges in model cascades.
2. There is a scalability issue regarding this method. Though it seems that only the small model would require extra fine-tuning. However, the labels of correct and incorrect are dependent on the specific large model. Namely, it will require a new round of fine-tuning if I switch to a new large model, which can be quite expensive, especially for generative models. Also, a very important knob here is $\alpha$, however, tuning that will require redo the finetuning phase. It sounds very suboptimal compared with the traditional methods for model cascades, which simply tune the threshold to control the deferrals.
3. The loss function seems too strong for fine-tuning small models, especially in generative models. As it would enforce the exact token matching, which might make it hard for the small model to converge?
4. Missing comparison to related works.
    4.1. For classification-related evaluation (Fig. 3), the baseline seems too weak. What’s the performance if I use a similar confidence calibration method in Fig. 5?
    4.2. I think the biggest motivation for the model cascade is the potential latency savings. How does this paper compare with speculative decoding in terms of latency savings? Note that speculative decoding can also apply fine-tuning to align the small model based on the large model.

    4.3 Other related works can be considered, for example, early exiting.

---

> ### Author Rebuttal · Authors · 2025-07-30
>
> We thank the reviewer for their thoughtful feedback and for highlighting the **clarity of our writing and the breadth of our evaluation**. We address the reviewer’s concerns regarding **novelty, scalability, and practical comparisons** below.
>
> ---
>
>
> ``1) Loss function has been studied and applied in previous works.``
>
> While we acknowledge that a similar loss has been proposed for OOD detection (e.g., Outlier Exposure), **our work repurposes this idea in a novel context**—namely, for deferral in model cascades. Unlike OOD detection, which typically relies on auxiliary datasets and disjoint distributions, deferral requires distinguishing between easy and hard in-distribution examples without additional supervision. Applying this loss to a standalone model would reduce utility by encouraging the model to "unlearn" difficult examples. However, in a cascade, **$\mathcal{M}_L$ recovers this lost performance, and $\mathcal{M}_S$ gains sharper confidence separation**, improving deferral quality without modifying its architecture or requiring access to $\mathcal{M}_L$. This use of the loss is specific to the cascade setting and enables practical gains in utility, latency, and deferral performance that prior work does not address. We have clarified this point in the paper.
>
> ---
>
> ``2) Scalability issues: labels of correct/incorrect depend on the large model; tuning \alpha requires redoing finetuning; suboptimal setup compared to traditional model cascades reliant on threshold tuning.``
>
> We thank the reviewer for raising the question of scalability. There are in fact two points of misunderstanding we would like to clarify.
>
> 1) **The “correct” versus “incorrect” labels used during fine-tuning come solely from the small model** evaluated on a labeled calibration dataset—not from the large model’s outputs (see Eqn. 1). In practice, one simply runs the small model on the fine-tuning set, compares its predictions to the true labels, and applies our Gatekeeper objective accordingly. Because this process does not depend on any remote model, **no additional fine-tuning is needed when you replace or upgrade the large model**.
>
> 2) The trade-off parameter $\alpha$ controls only how strongly the loss encourages low confidence on misclassified examples versus high confidence on correct ones; it **does not affect the gating threshold used at inference time**. As detailed in Stage 3 (see Eqn. 3), deferral decisions remain based on a separate confidence or entropy threshold, which can be tuned cheaply at deployment without retraining. We agree, however, that if one wishes to explore a different $\alpha$ to shift the calibration–accuracy trade-off, a new fine-tuning run of the small model would be required. As we note in lines 183–193, this flexibility enables practitioners to select the desired balance between conservatism in uncertain regions and decisiveness on easy examples, tailored to the cost and performance constraints of their specific application.
>
> ---
>
> ``3) Loss function seems too aggressive for fine-tuning small models in generative setups.``
>
> We agree with the reviewer that **Gatekeeper can be overly aggressive in generative settings where multiple token sequences may convey the same meaning**. As already noted in our limitations (lines 366–369), relying on exact token matches may over-penalize semantically valid outputs. While recent approaches such as **semantic entropy [1] aim to address this, they typically require querying a larger model to judge semantic correctness**—effectively constituting a deferral, which defeats the purpose of our cascade setup (see lines 131–137 where we already elaborate on this point). To the best of our knowledge, there is currently no lightweight alternative that enables small models to assess semantic correctness without external assistance. Nevertheless, despite this limitation, **our results in Section 4.3 (e.g., Figure 7) show consistent improvements in deferral performance for both LM and VLM tasks**, suggesting that even token-level confidence calibration can meaningfully improve reliability in generative setups.
>
> [1] Kuhn, Lorenz, Yarin Gal, and Sebastian Farquhar. "Semantic uncertainty: Linguistic invariances for uncertainty estimation in natural language generation." ICLR 2023.
>
> ---
>
> ``4.1) For classification-related evaluation (Fig. 3), the baseline seems too weak. What’s the performance if I use a similar confidence calibration method in Fig. 5?``
>
> We thank the reviewer for the question but believe there may be a misunderstanding: **Figure 3 is purely illustrative and does not report any empirical results**—it serves to explain the evaluation metric used throughout the paper. The actual classification and language model experiments appear in **Figures 4 and 5, which both include an untuned baseline and strong cascading methods [1,2]. Figure 5 additionally includes prompting-based baselines [3] that are only applicable to language models and cannot be transferred to image classification tasks**. We kindly ask the reviewer to follow up with us in case we misunderstood their concern.
>
> [1]: Narasimhan, Harikrishna, et al. "Post-hoc estimators for learning to defer to an expert." NeurIPS 2022.
>
> [2]: Gupta, Neha, et al. "Language model cascades: Token-level uncertainty and beyond." ICLR 2024.
>
> [3]: Kadavath, Saurav, et al. "Language models (mostly) know what they know." arXiv preprint arXiv:2207.05221 (2022).
>
> ---
>
> ``4.2) How does this paper compare with speculative decoding in terms of latency savings?``
>
> We appreciate the reviewer’s interest in speculative decoding, yet we respectfully believe that a direct latency comparison would be misleading:
>
> 1) Speculative decoding accelerates every input by asking the large model to verify draft tokens that the small model proposes, so **the large model is still invoked on all inputs**. Our cascade, by contrast, aims to avoid calling the large model on inputs the small model can already solve. Consequently, speculative decoding optimizes in-place token-level latency, while our method optimizes end-to-end compute and monetary cost at the request level. Because **the two techniques improve fundamentally different bottlenecks**, a like-for-like timing chart would conflate distinct operating points.
>
> 2) As noted in the introduction and our related work section, **the two approaches can be composed**: after our deferral gate decides to consult the large model, one could still decode that portion of the input speculatively (see [1] for an example of such hybrids). The proper question is therefore not *“Which one is faster?”* but “How much additional speed-up does deferral enable on top of speculative decoding?” Measuring that joint benefit would require re-engineering both systems into a single pipeline, which is orthogonal to the paper’s core contribution—namely, a loss that teaches the small model when to defer.
>
> We have added an extended discussion on these notable differences/similarities as part of Appendix §B.1.
>
> [1] Narasimhan, Harikrishna, et al. "Faster cascades via speculative decoding." ICLR 2025.
>
> ---
>
> ``4.3) Relationship to early exiting.``
>
> We appreciate the suggestion to include early-exit experimentation, but we believe that—similar to the speculative decoding angle—an empirical comparison with early–exiting methods would add little insight for three reasons:
>
> 1) Early–exiting aims to *skip* the remaining layers of a single network once an intermediate classifier is sufficiently confident, thereby offering a **layer–level latency–accuracy trade-off** within *one* model. **Our work tackles the fundamentally different problem of *model–level* deferral between a *small* and a *large* model**. The goal is to keep the large model entirely idle for easy requests and to invoke it only when needed. Hence our primary measure of success is *joint accuracy versus cross-model compute budget*, not marginal delay per layer inside a fixed backbone. Adding an early-exit baseline would therefore **conflate two orthogonal trade-offs** and obscure the key question we study: *How well can a calibrated small model decide when to hand off to a more capable expert?*
>
> 2) Early–exit usually presupposes **white-box control over the entire network architecture** so that branch classifiers can be inserted and jointly trained (e.g., BranchyNet, Adaptive Neural Networks). Our cascade setting explicitly targets **heterogeneous or API-based experts**—large language models, vision–language models, and so forth—that cannot be modified. Early–exits cannot be applied in such black-box situations, whereas our \loss fine-tuning remains feasible. Moreover, early–exiting has seen limited success for *autoregressive* sequence generation, where every token depends on the full hidden state; in contrast, our experiments span both classification and open-ended generation tasks.
>
> 3) Because early–exit and cascading operate at different granularities, they are **complementary rather than competing** techniques. One could in principle insert early exits *inside* the small model and still rely on calibrated deferral to the large model—yielding a three-level system (exit-1 → exit-2 → large model). Evaluating that combined design is a compelling direction for future engineering work, but it lies beyond the scope of our current study, whose contribution is a general-purpose confidence-tuning loss independent of architectural changes.
>
> Again, we have added an extended discussion to Appendix §B.1.
>
> [1] Teerapittayanon, Surat, et al. "Branchynet: Fast inference via early exiting from deep neural networks." ICPR 2016.
>
> [2] Bolukbasi, Tolga, et al. "Adaptive neural networks for efficient inference." ICML 2017.
>
> ---
>
> **We hope that our rebuttal has clarified the reviewer’s questions and concerns and that the reviewer considers raising their individual and overall score(s). We are happy to further engage with them during the discussion period should further questions arise!**

---

> > ### Comment · Reviewer_aLX9 · 2025-07-31
> > **response**
> >
> > Thank you for the detailed explanation—it addresses some of my concerns. However, I still feel uncertain about the fine-tuning phase. Relying solely on the small model for fine-tuning still seems suboptimal to me. If the input is particularly challenging, neither the small nor the large model may be able to solve it. In such cases, should the input be routed to the large model or not? Only involving the large model during calibration seems to address this class of difficult inputs effectively.

---

> > > ### Author Response · Authors · 2025-08-01
> > > **Clarification on universally hard-to-classify data points**
> > >
> > > We thank the reviewer for their swift reply to our rebuttal! We appreciate the reviewer pointing out the difficulty of dealing with examples that are inherently hard to classify for both models. **We would like to alert the reviewer to our reply to Reviewer 5Nnu (points 1, 2 and 3), where we discuss a similar concern**. We extend on this discussion below.
> > >
> > > **Behavior on “unsolvable” inputs**: First, we want to clarify that even when both models will err our deferral strategy remains safe:
> > > - The small model’s confidence after **Gatekeeper fine-tuning is strictly lower on such hard inputs**; hence they are routed upward by Equation (3).
> > > - While the large model may still fail, **cascading cannot degrade performance relative to accepting the small model**, because we assumed (and empirically confirmed) that the large model is at least as accurate as the small model. In other words, the worst-case risk of our cascade is bounded by the large model’s risk, while its average cost is often far lower.
> > >
> > > **The challenge of three-way decisions**: To the best of our understanding, the reviewer is effectively asking for a 3-way decision problem:
> > >
> > > - **accept local**: $\mathcal{M}_S$ can correctly handle the data point without needing any assistance from $\mathcal{M}_L$.
> > > - **defer to remote**: $\mathcal{M}_S$ can not correctly handle the data point but $\mathcal{M}_L$ can.
> > > - **reject local**: neither $\mathcal{M}_S$ nor $\mathcal{M}_L$ can correctly handle the data point.
> > >
> > > While we agree that this is a desirable modeling assumption to optimize compute and communication overhead, there are significant challenges associated with this formulation:
> > >
> > > 1) **Black-box or volatile large models**. In many commercial settings the large model is an external API whose weights, failure modes, and update frequency are opaque or hard to estimate. Teaching the small model to estimate the large model’s uncertainty would therefore **(i) require querying the large model extensively** during fine-tuning, inflating cost and latency, and **(ii) become stale the moment the provider upgrades the large model**. For example, it is possible that errors made by a previously used large model were not due to irreducible label noise but instead due to limited approximation quality or poor training. Had we instructed $\mathcal{M}_S$ to reject these instances, we would then have to retrain $\mathcal{M}_S$ to account for a better remote model. By relying solely on signals that $\mathcal{M}_S$ can generate, **our method stays provider- and version-agnostic**. We believe that this is a key advantage of our method.
> > >
> > > 2) **Large error surface saturates small model capacity**. Predicting its own errors already taxes a small model; **predicting the intersection of its errors and the large model’s errors is far harder**. In practice this means approximating a highly irregular, high-dimensional decision boundary that reflects (i) the small model’s representational limits, (ii) the large model’s vastly richer hypothesis class, and (iii) the unknown correlations between their mistakes. Fitting such structure would require the small model to allocate parameters to regions of input space it never has to classify itself—potentially wasting capacity that should be devoted to its primary task. Gatekeeper side-steps this by focusing solely on patterns the small model can reliably sense: when it is right versus wrong.
> > >
> > > 3) **Architectural challenges**. A three-way scheme (accept, defer, reject) **converts the gating problem from a 1-D threshold to a multi-class classification head**. In vanilla softmax architectures the only scalar readily available at inference is predictive entropy (or max-softmax). Entropy already captures the pattern “low confidence → defer,” but it offers no additional signal to cleanly separate “defer” from “reject.” **Implementing a reliable “reject” branch therefore demands a redesigned rejection head** trained with custom targets (adding parameters, latency, and a second hyper-parameter threshold). Such a process would likely need to tune the small model first, freeze it, add the reject classifier, and tune it separately. Conversely, by retaining the single-scalar entropy gate **we preserve the properties practitioners care about**: one tunable threshold, monotone risk–coverage behavior, and straightforward post-deployment monitoring. Gatekeeper thus plugs into any existing small model without modifying its computation graph, while solutions that seek local rejection require a redesigned network and a more complex calibration pipeline.
> > >
> > > We have incorporated this discussion into the **revised Limitations paragraph** and added these points to the appendix to clarify why extending to a three-way gating mechanism—while appealing in principle—lies beyond the practical and methodological scope of the present work.
> > >
> > > ---
> > >
> > > **We hope that our response has clarified the reviewer’s concern and continue to be available for further clarification requests.**

---

### Official Review · Reviewer_Eqgu · 2025-07-03

**Clarity:** 2
**Significance:** 2
**Originality:** 2
**Rating:** 3
**Confidence:** 3

**Summary:**

The authors introduce Gatekeeper, a novel loss function designed to improve the performance of small models in model cascade systems by tuning their confidence calibration. The method enables the small model to identify when to defer to a more capable large model, aiming to optimize deferral performance without modifying the architecture or requiring auxiliary deferral modules. Gatekeeper encourages high confidence on correct predictions and uniform uncertainty on incorrect ones through a linear combination of a cross-entropy and a KL-divergence loss. The approach is validated on a wide range of tasks (image classification, language modeling, and vision-language tasks ) and architectures (encoder-only, decoder-only, and encoder-decoder models). The results show notable improvements in deferral metrics over baselines.

**Questions:**

See weakness in loss implementation

**Ethical Concerns:**

["NO or VERY MINOR ethics concerns only"]

**Final Justification:**

I appreciate the authors' efforts in the rebuttal. The further comparison with previous work and explanation of the implementation details were helpful. I have increased my score accordingly.

**Limitations:**

No specific potential negative societal impact

**Paper Formatting Concerns:**

No specific concern

**Quality:**

2

**Strengths And Weaknesses:**

### **Strengths**

1. **Novel adaptation of a hybrid loss for cascading systems**: The introduction of a loss function that penalizes incorrect predictions with uniform confidence is novel in the context of model cascades.
2. **Strong empirical validation**: The method is tested across diverse tasks and model architectures, demonstrating robust improvements in deferral performance.

### **Weaknesses**

**1. Limited exploration of alternative confidence metrics and loss formulations:**
While the authors frame the use of the Gatekeeper loss as a novel contribution to the cascading setup, the core idea—using confidence calibration to distinguish correct from incorrect predictions—has been extensively studied in the broader uncertainty estimation literature. As such, the novelty of adapting this loss to cascaded inference may be limited. To strengthen the empirical contribution, the authors should include a broader comparison against alternative loss functions and calibration techniques from prior work. I highly suggest authors to do a more through literature search on these domains (e.g., start with [A][B][C] papers and check papers that cite these works) to better contextualize the proposed approach and clarify its advantages in model cascading setup.

**2. Ambiguity in loss computation details:** The paper lacks clarity regarding the implementation of the Gatekeeper loss (Eq. 1, Lines 172–193). It remains unclear how the model determines which samples are incorrect predictions, and whether the loss terms $L_{\text{corr}}$ and $L_{\text{incorr}}$ are computed in a single forward pass or through separate evaluation and sampling steps. For instance, do the authors generate separate batches for correct and incorrect samples after a full forward pass, or do they compute both terms within each mini-batch during training? Without a clear explanation, it is difficult to reproduce the training pipeline or assess the computational overhead. This should be clarified with an explicit algorithmic description or pseudocode.

---

> ### Author Rebuttal · Authors · 2025-07-30
>
> We thank the reviewer for recognizing the **novelty of our loss in the cascading setting and the strength of our empirical results**. We appreciate the constructive **suggestions regarding broader comparisons and implementation clarity**, and address both issues below:
>
> ---
>
> ``1)  Limited exploration of alternative confidence metrics and loss formulations.``
>
> We appreciate the reviewer’s observation that confidence calibration has a long history and agree that situating Gatekeeper more explicitly within that literature will clarify its added value for cascaded inference. However, **the reviewer did unfortunately not specify what the listed references [A][B][C] are** (paper names and authors are not provided in the review). As a result, we were unable to accommodate any additional experimentation during the initial rebuttal. Nevertheless, **we have added an additional section in the Appendix (§B.4) that now surveys canonical calibration objectives**—including temperature scaling, focal loss, confidence penalty, and outlier-exposure style uniformity losses—and explains how Gatekeeper extends these ideas by *jointly* sharpening correct predictions *and* flattening incorrect ones so that a single scalar confidence score yields a clean separation for deferral. These additions, together with our existing comparisons to cascading methods [1,2], and prompting baselines [3], demonstrate that Gatekeeper provides a simple yet consistently stronger solution for confidence-based deferral, thereby addressing the reviewer’s concern about novelty and empirical breadth.
>
> We include a shortened version of our new appendix section below:
>
> > ### C.4 Gatekeeper in the Context of Canonical Calibration Objectives
> >
> >**Motivation.**  Section 4 showed that the Gatekeeper loss improves *deferral performance* with minimal implementation effort. Because many calibration objectives also manipulate confidence, we now position Gatekeeper relative to four widely–used losses.
> >
> > **Canonical calibration objectives.**  Let $p_{\theta}(y\mid\mathbf{x})$ denote the softmax (or token) distribution predicted by a model with parameters $\theta$, and let $y^\star$ be the ground-truth label.
> >
> >1. **Temperature scaling** [4]:
> >   Applies a single scalar $T{>}0$ at test time: $p_T(y\mid\mathbf{x}) \propto \exp(z_c/T)$. It preserves the rank ordering and therefore *cannot tighten* the ranking-based risk–coverage curve, but can improve threshold-based acceptance.
> >
> >2. **Focal loss** [5]:
> >   Adds a down-weighting factor to easy examples, $\text{FL}(p,y^\star)=-(1-p_{y^\star})^\gamma \log p_{y^\star}$ with $\gamma \in [0,\infty)$. It improves class imbalance calibration but does not explicitly penalize over-confidence on *incorrect* samples.
> >
> >3. **Confidence penalty** [6]:
> >   Regularises high-entropy predictions through $\text{CE}(p,y^\star) + \lambda \mathcal{H}(p)$. While it flattens *all* distributions, it does not distinguish between correct and incorrect cases.
> >
> >4. **Outlier exposure (OE)** [7]:
> >   Adds a KL-to-uniform loss on auxiliary **OOD** data, mirroring the second term of Gatekeeper but only on outliers, not in-distribution misclassifications.
> >
> >**How Gatekeeper differs.**  **Gatekeeper** combines *two complementary gradients*:
> >- (i) a standard CE term on correct predictions with an instance-level mask, thereby **sharpening** those logits;
> >- (ii) a KL-to-uniform term on *incorrect* predictions, **flattening** their confidence.
> >
> >This asymmetric design forces the scalar summary $g(\mathbf{x}) = \max_c p_\theta(y = c \mid \mathbf{x})$ (or token-entropy) to separate correct from incorrect points *without* requiring additional heads, auxiliary datasets, or test-time tuning.
>
>
> **References**:
>
> [1]: Narasimhan, Harikrishna, et al. "Post-hoc estimators for learning to defer to an expert." Advances in Neural Information Processing Systems 35 (2022): 29292-29304.
>
> [2]: Gupta, Neha, et al. "Language model cascades: Token-level uncertainty and beyond." ICLR 2024.
>
> [3]: Kadavath, Saurav, et al. "Language models (mostly) know what they know." arXiv preprint arXiv:2207.05221 (2022).
>
> [4]: Guo, Chuan, et al. "On calibration of modern neural networks." ICML 2017.
>
> [5]: Lin, Tsung-Yi, et al. "Focal loss for dense object detection." ICCV 2017.
>
> [6]: Pereyra, Gabriel, et al. "Regularizing neural networks by penalizing confident output distributions." arXiv preprint arXiv:1701.06548 (2017).
>
> [7]: Hendrycks, Dan, Mantas Mazeika, and Thomas Dietterich. "Deep anomaly detection with outlier exposure." ICLR 2019.
>
> ---
>
> ``2) Ambiguity in loss computation details.``
>
> We agree with the reviewer that our original paper draft did not explain the loss computation in sufficient detail. As a result, **we have added the following clarifying paragraph to our current draft**.
>
> >**Practical computation of the Gatekeeper loss**. We compute the Gatekeeper loss *once per mini-batch* within the standard training loop—no auxiliary passes or data-set re-shuffling are required. Given a mini-batch $B=\\{ (x_i, y_i)\\}_{i=1}^{N}$, we perform a single forward pass through $\mathcal{M}_S$. This allows us to obtain
> >
> >- a logit vector $z_i=f_\theta(\mathbf{x}_i)$ and
> >- predicted labels $\hat{y_{i}} = \arg \max_{c} z_{i,c}$.
> >
> >Two binary masks $m^{\text{corr}}_i=I\\{y_i=\hat{y}_i\\}$ and $m^{\text{incorr}}_i= \neg m^{\text{corr}}_i$ are computed *on-the-fly*. The hybrid loss is then assembled in a fully vectorized manner:
> >
> >$$
> >\mathcal{L}_\text{corr} = \frac{1}{N}\sum_i m^{\text{corr}}_i \operatorname{CE} (p_i,y_i)
> >$$
> >
> >$$
> >\mathcal{L}_\text{incorr} = \frac{1}{N}\sum_i m^{\text{incorr}}_i \operatorname{KL} (p_i | | \mathcal{U})
> >$$
> >
> >where $p_i=\operatorname{softmax}(z_i)$. Because both masks and losses are computed inside the same tensor graph, back-propagation incurs only the cost of $\mathcal{O}(N \times C)$ element-wise operations—identical to a vanilla cross-entropy step. This single-pass design keeps the computational overhead negligible while guaranteeing that in the full loss $\mathcal{L} = \alpha \mathcal{L}\_\text{corr} + (1 - \alpha) \mathcal{L}\_\text{incorr}$ every sample contributes to either $\mathcal{L}\_\text{corr}$ or $\mathcal{L}\_\text{incorr}$ in the same optimization step.
>
> ---
>
> **We hope that our rebuttal has addressed the reviewer’s questions and concerns and that the reviewer considers raising their individual and overall score(s). Our work has already noticeably improved thanks to their feedback. We are happy to further engage with them during the discussion period!**

---

> > ### Comment · Reviewer_Eqgu · 2025-08-03
> >
> > I appreciate the authors' efforts in the rebuttal. The further comparison with previous work and explanation of the implementation details were helpful. I have increased my score accordingly.

---

> ### Comment · Reviewer_Eqgu · 2025-08-03
> **Updates on the A/B/C references**
>
> Sorry, I found that the A/B/C references in the original review were missed when I was copying and pasting from the local draft. Here they are:
>
> [A] Szegedy et al., Rethinking the Inception Architecture for Computer Vision, CVPR 2016
>
> [B] Pereyra et al., Regularizing Neural Networks by Penalizing Confident Output Distributions, ICLR 2017
>
> [C] DeVries and Taylor, Learning Confidence for Out-of-Distribution Detection in Neural Networks, 2018

---

> > ### Author Response · Authors · 2025-08-05
> > **Additional Discussion on Related Work**
> >
> > We sincerely thank the reviewer for providing these additional references, which we have now carefully examined. We discuss our insights below:
> >
> > ---
> >
> > **[A] Szegedy et al., Rethinking the Inception Architecture for Computer Vision, CVPR 2016:**
> > This paper proposes architectural refinements to convolutional neural networks, most notably the use of Inception modules and auxiliary classifiers to improve optimization and generalization. While auxiliary heads are introduced, their role is strictly limited to acting as regularizers during training to help with convergence. *The paper does not consider abstention, deferral, or selective prediction*, nor does it explore how confidence estimates might be used to inform downstream decisions. As such, we do not believe this paper provides relevant insight for designing confidence-based deferral mechanisms in cascaded systems.
> >
> > **[B] Pereyra et al., Regularizing Neural Networks by Penalizing Confident Output Distributions, ICLR 2017:**
> > This work is more closely aligned with our interest in confidence estimation. The authors propose an entropy-based regularizer that penalizes low-entropy (i.e., overconfident) predictions to improve generalization. *While this technique shapes the confidence landscape of the model, it is developed entirely in the context of improving performance on the primary task under the i.i.d. training distribution*. The paper does not consider how confidence might be used as a decision signal for abstention or deferral, nor does it evaluate its approach in any selective prediction or cascading setting.
> >
> > **[C] DeVries and Taylor, Learning Confidence for Out-of-Distribution Detection in Neural Networks, 2018:**
> > Among the three works, *this paper is the most related to our setting*. It proposes training an auxiliary confidence head alongside the main classifier, with a penalty that encourages the model to assign low confidence to difficult or unfamiliar examples. This score is then thresholded at test time to abstain from uncertain predictions, enabling selective prediction for OOD detection. While the original application does not involve a second-stage model, the method provides a natural precursor to cascading setups where uncertainty governs deferral. In fact, *the “Post-hoc Estimators for Learning to Defer to an Expert” method [1], a baseline which we already compare against in our experimental panel, can be viewed as an explicit adaptation of this idea for cascading systems*: it uses a separate deferral head trained post-hoc to decide when to defer to a large expert. Because [1] builds directly on the core idea introduced in [C], we believe an explicit empirical comparison to [C] is not necessary and effectively provided via [1]. However, due to the conceptual similarities between these two works, *we now explicitly mention [C] in our extended related work section (Appendix B.1) and discuss the similarities and differences between [C] and [1]*.
> >
> > [1]: Narasimhan, Harikrishna, et al. "Post-hoc estimators for learning to defer to an expert." Advances in Neural Information Processing Systems 35 (2022): 29292-29304.
> >
> > ---
> >
> > **We hope that our response has clarified the reviewer’s concern and continue to be available for further clarification requests.**

---

### Official Review · Reviewer_ri54 · 2025-07-04

**Clarity:** 4
**Significance:** 2
**Originality:** 3
**Rating:** 5
**Confidence:** 3

**Summary:**

The paper introduces the “GATEKEEPER” loss, which fine-tunes a lightweight model to escalate hard-to-label samples to a larger, more capable model. The objective is to balance overall accuracy while minimizing queries to the computationally heavier model. Experiments validate the effectiveness of GATEKEEPER across classification, language, and vision-language tasks.

**Questions:**

Q1. Would GATEKEEPER scale beyond two stages?
 In other words, could you train a $k$-level cascade: $\mathcal{M}_1 \rightarrow \mathcal{M}_2 \rightarrow … \rightarrow \mathcal{M}_K$ with increasingly large models and apply the loss hierarchically?


Q2. Could you also explain why you regard the assumption that “$\mathcal{M_S}$ makes the same mistakes as $\mathcal{M}_L$” as requisite for your method development? L139-L140

**Ethical Concerns:**

["NO or VERY MINOR ethics concerns only"]

**Final Justification:**

I increased my rating to "accept" as my major concerns have been adressed during the rebuttal.

**Limitations:**

Oracle scenario omitted.

The current study always treats a neural model as the “large” backend. An informative upper-bound experiment would replace the backend with an Oracle (e.g., ground-truth label or a human expert) and treat today’s large model as the new $\mathcal{M}_S$.

This would strengthen the study by revealing how much of the joint-accuracy gain stems from better routing versus from the back-end’s inherent power and would inform human-in-the-loop deployments of autonomous systems where server cost is secondary to trusting the system’s outputs. This should be feasible with a simple simulation and no extra data collection.

**Paper Formatting Concerns:**

In the Methods section, “Stage 1” is not an algorithmic step. It actually states the prerequisite (having a pre-trained small model). As written, readers may assume it is part of the GATEKEEPER procedure. Consider re-labelling this block as “Prerequisite” (or folding it into a short preamble) and renumbering the subsequent block as “Stage 1: Confidence-tuning with GATEKEEPER.” This small change would make the true start of the method immediately clears [Adressed]

**Quality:**

3

**Strengths And Weaknesses:**

Strengths
- Well-written paper and relevant appendices.
- The method supports heterogeneous architectures in the model pair.
- Comprehensive related work and a well-positioned contribution.
- Extensive experiments.
- Flexible setup via the α-parameter: one can tune the system to query the large model for high-risk applications (e.g., medical) and lean more toward the edge model for lower-risk scenarios (e.g., traffic).

Weaknesses:

1. Several assumptions may not hold in real settings and are not verified:
  - The same mistakes made by $\mathcal{M}_L$ would be made by $\mathcal{M}_S$), especially with heterogeneous architectures.
  - The method assumes Acc($\mathcal{M}_L$) > Acc($\mathcal{M}_S$). This can fail after fine-tuning, particularly on a restricted test domain, where studies show Acc($\mathcal{M}_S$ fine-tuned on $X$) > Acc($\mathcal{M}_L$ on $X$).

2. It is unclear whether higher joint accuracy results from a genuine deferral process or simply from fine-tuning on the test domain.

3. The study does not assess the case where $\mathcal{M}_S$ is fine-tuned on pseudo-labels from $\mathcal{M}_L$ in a test-time adaptation setting.

---

> ### Author Rebuttal · Authors · 2025-07-30
>
> We thank the reviewer for their thoughtful and positive assessment, including their **recognition of our clear writing, comprehensive experiments, and practical, flexible design**. They raise important **concerns about real-world assumptions—such as model agreement, post-fine-tuning accuracy, and pseudo-labeling**—which we address below:
>
> ---
>
> ``W.1 + Q.2) Why is the assumption “M_S makes the same mistakes as M_L” necessary?.``
>
> The premise that every error made by the large expert model $\mathcal{M}_L$ is also made by the small model $\mathcal{M}_S$ **follows directly from well-documented scaling-law behavior**: when two models are trained on comparable data with the same objective, predictive accuracy increases monotonically with parameter count and compute budget. Empirically, larger vision transformers and language models dominate their smaller counterparts across virtually every benchmark, a pattern quantified by [1] and many subsequent works. Embracing this monotonicity gives us two critical benefits.
> 1) **It lets us treat $\mathcal{M}_L$ as an *upper envelope* of capability**: deferring from $\mathcal{M}_S$ to $\mathcal{M}_L$ can only maintain—or improve—performance, never degrade it. Without that guarantee, the deferral problem becomes ill-posed because sending an input to $\mathcal{M}_L$ could occasionally lower accuracy—forcing the gate to reason about *relative* per-example competence instead of simple uncertainty in $\mathcal{M}_S$.
> 2) **The assumption is baked into our evaluation protocol and the definition of the ideal deferral curve against which we compute the deferral-performance score $s_d$**. That oracle curve assumes perfect identification of samples misclassified by $\mathcal{M}_S$ *and* subsequent perfect correction by $\mathcal{M}_L$; it therefore constitutes a true upper bound only when $\mathcal{M}_L$’s error set is a subset of $\mathcal{M}_S$’s. If that subset relation does not hold, any metric normalised by the oracle curve would be inflated in expectation, obscuring real differences between methods.
>
> In short, the **assumption is not an arbitrary convenience**—it is a principled reflection of established scaling trends that (i) shields deferral from pathological failure cases and (ii) grounds our performance metric in a meaningful, attainable ceiling. We are happy to expand on this in our updated paper draft.
>
> [1] Kaplan, Jared, et al. "Scaling laws for neural language models." arXiv preprint arXiv:2001.08361 (2020).
>
> ---
>
> ``W.2) It is possible that Acc(M_S fine-tuned on X) > Acc(M_L on X). Also, it is unclear whether higher joint accuracy results from a good deferral or simply from fine-tuning on the test domain.``
>
> This is a valid concern. However, in our experimentation, **we ensure that both the small and large models are always trained or fine-tuned on the same training split of the target dataset before we apply Gatekeeper to the small model only**. Our fine-tuning stage with Gatekeeper alters only the confidence profile of the small model—its class-probability outputs—not the underlying data distribution, and the large model remains completely frozen. Consequently, whenever joint accuracy improves while the small model’s stand-alone accuracy remains constant or even drops, **the only mechanism that can explain the gain is more effective routing of hard examples to the large model** (which is precisely what we observe; see the rightmost panel in Figures 4 and 5 showing noticeably better performance for $\mathcal{M}_L$ over $\mathcal{M}_S$). Moreover, **all baselines are tuned under the same protocol**, so the relative improvements we report isolate the contribution of the deferral strategy itself, not differences in domain exposure or training budget. We have clarified this in our experimental setup paragraphs in Section 4.
>
> ---
>
>
> ``W.3) What if M_S is fine-tuned on pseudo-labels from M_L?``
>
> We agree that one could continually tune $\mathcal{M}_S$ on pseudo-labels from $\mathcal{M}_L$, but **this falls outside the deployment regime we study**. In our setting **(i)** $\mathcal{M}_L$ is an immutable resource beyond the control of the user (after initial fine-tuning); and **(ii)** the deferral gate must decide whether to call $\mathcal{M}_L$ without first querying it, because every additional call already incurs the latency and cost we seek to avoid. Streaming logits or hard labels from $\mathcal{M}_L$ for ongoing adaptation would therefore nullify the efficiency gains of deferral. **Gatekeeper is, however, agnostic to how the initial $\mathcal{M}_S$ was obtained**: if practitioners start with an $\mathcal{M}_S$ distilled from $\mathcal{M}_L$ (like we often do in our experiments), they can still apply Gatekeeper to sharpen the confidence gap needed for reliable deferral—without any extra expert queries at inference time. Fully exploring that hybrid pipeline is an interesting avenue for future work, but it involves distinct engineering and economic trade-offs and would require a broader experimental setup than what fits in the present study.
>
> ---
>
> ``Q.1) Scaling Gatekeeper to multi-level cascades M_1 → M_2 → … → M_K.``
>
> This is indeed an interesting extension. **Gatekeeper makes no explicit assumption on the number of cascade levels**—it is inherently modular and agnostic to model architecture. In practice, one can **fine‐tune each model $\mathcal{M}_i$ in the chain independently using the Gatekeeper loss with a stage‐specific hyperparameter $\alpha_i$** that governs the balance between confident acceptance and deferral. A natural hierarchical training procedure is to first apply Gatekeeper to $\mathcal{M}_1$ to calibrate its confidence, then apply it to $\mathcal{M}_2$ on the subset of inputs deferred by $\mathcal{M}_1$, and so on through $\mathcal{M}_K$. This approach scales linearly in the number of levels and incurs only the familiar overhead of per‐model fine‐tuning—no joint optimization or additional routing networks are required. We therefore expect that deeper cascades will offer flexible trade‐offs between computational cost and predictive accuracy across multiple model tiers, and we leave a full empirical evaluation of $K>2$ cascades to future work.
>
> ---
>
> ``Replace big model with a perfect label oracle.``
>
> We appreciate the reviewer’s interest in an oracle-backed simulation; however, **replacing the back-end with ground-truth labels would shift the task from *cascading* to *selective classification***, where the objective is abstention instead of deferral. Our goal in this paper is noticeably different: we study how to *optimize the small model’s confidence* so that it routes inputs judiciously when the only fallback is a *fixed, imperfect* but highly capable model—exactly the constraint faced in real-world cascades where API calls or server latency make further human verification impractical. To keep the exposition focused, we therefore do *not* evaluate deferral quality via the risk–coverage framework but instead **resort to the normalized area metric $s_d$ (Eq. 7)**, which already benchmarks each method against its own *ideal* routing curve; thus the performance gap we report isolates routing effectiveness from raw back-end accuracy. **Performance evaluation of selective classifiers [1,2,3,4] as well as oracle-style upper bounds for rejection [5,6] on the other hand have been thoroughly explored in prior SC work**, so duplicating them here would add length without clarifying our contribution. Nevertheless, we will note this distinction in a separate appendix section in the revision and cite the relevant SC papers to guide readers interested in SC-related oracle analyses.
>
> [1] Geifman, Yonatan, and Ran El-Yaniv. "Selectivenet: A deep neural network with an integrated reject option." International conference on machine learning. PMLR, 2019.
>
> [2] Huang, Lang, Chao Zhang, and Hongyang Zhang. "Self-adaptive training: beyond empirical risk minimization." Advances in neural information processing systems 33 (2020): 19365-19376.
>
> [3] Liu, Ziyin, et al. "Deep gamblers: Learning to abstain with portfolio theory." Advances in Neural Information Processing Systems 32 (2019).
>
> [4] Jiang, Heinrich, et al. "To trust or not to trust a classifier." Advances in neural information processing systems 31 (2018).
>
> [5] Geifman, Yonatan, Guy Uziel, and Ran El-Yaniv. "Bias-reduced uncertainty estimation for deep neural classifiers." arXiv preprint arXiv:1805.08206 (2018).
>
> [6] Rabanser, Stephan, et al. "Training private models that know what they don’t know." Advances in Neural Information Processing Systems 36 (2023): 53711-53727.
>
> ---
>
> ``Re-label Stage 1 as a prerequisite``
>
> This is a good suggestion; indeed Stage 1 is not really a part of the Gatekeeper tuning process itself! We have restructured this part of the paper to **state Stage 1 as a preliminary setup step**. With this change Stage 2 (now Stage 1) now corresponds to the Gatekeeper training stage and Stage 3 (now Stage 2) now corresponds to the deployment stage.
>
> ---
>
> **We hope that our rebuttal has addressed the reviewer’s questions and concerns and that the reviewer considers raising their individual and overall score(s). Our work has already noticeably improved thanks to their feedback. We are happy to further engage with them during the discussion period!**

---

> > ### Comment · Reviewer_ri54 · 2025-08-04
> > **Answer to Rebuttal**
> >
> > Thank you for the answers. There remains one point to discuss:
> >
> > Claim. “$M_S$ makes the same mistakes as $M_L$,” can be interpreted that the error–set inclusion
> > $E_L \subseteq E_S$ is valid.
> >
> > This is incorrect. I provide a counterexample with YOLOv11 and RT-DETR
> > showing instances where the teacher model have mistakes that the student do not have.
> > Link: \url{https://wormhole.app/AYq57Z#slspWDvLA-9FBZYxlkH96A}. Expire after 24 hour.
> >
> > Request. Please revise the assumption to the weaker statement that the teacher
> > has higher \emph{average} predictive power than the student, e.g.
> >
> > $\mathrm{Acc}(M_L) \ge \mathrm{Acc}(M_S)$
> >
> > Or let me know if I have not understood the issue.

---

> ### Author Response · Authors · 2025-08-04
> **Further Clarification on Our Dominance Assumption**
>
> We thank the reviewer for their engagement in the discussion phase and for sharing their remaining concerns with us! We provide **additional clarification on our dominance assumption** below:
>
> ---
>
> **Clarification of scope**: We want to clarify that **our claim was never intended as a universal law**. As the reviewer rightly notes—and as we acknowledged in our original rebuttal—a smaller model can outperform a larger one on individual inputs under several well-studied conditions, including but not limited to: distribution shift, aggressive domain-specific fine-tuning, adversarial robustness artifacts, differing architectural inductive biases (e.g., convolutional versus transformer backbones), heavy regularization or pruning applied only to the large model, and data sparsity in the large model’s training regime.
>
> **Empirical validity within our experiments**: Across all datasets, seeds, and model pairs in our experimental suite, **we did not observe any instances where the small model was correct while the large model was incorrect**. Thus the strict error-set inclusion $E_L \subseteq E_S$​ holds for our results, and the corresponding oracle deferral curve remains a valid upper bound.
>
> **Revision of the paper’s statement**: Reconciling these two insights and recognizing that practitioners wanting to implement Gatekeeper might not necessarily observe the same empirical behavior if they use different models or data sets (including the example shared by the reviewer), **we are implementing a precise definition of our with-high-probability dominance assumption in the paper**. Our previous draft stated in lines 139-141:
>
> > As part of our setup, we assume that $\mathcal{M}\_S$ is strictly less capable than $\mathcal{M}\_L$ --- a realistic scenario in practice supported by scaling laws (Kaplan et al., 2020). Under this assumption, mistakes made by $\mathcal{M}\_L$ are also made by $\mathcal{M}\_S$; however, $\mathcal{M}\_S$ may make additional errors that $\mathcal{M}\_L$ would avoid. This reflects the general observation that larger models tend to outperform smaller models across a wide range of tasks.
>
> Our updated version now reads as follows:
>
> > As part of our setup, we assume that $\mathcal{M}\_S$ dominates $\mathcal{M}\_L$ as per the following assumption.
> >
> >
> > **Dominance assumption.**
> > Let $\mathcal{D}$ denote the target deployment distribution. We assume that $\mathcal{M}\_L$ dominates the $\mathcal{M}\_S$ *with high probability* under $\mathcal{D}$; formally,
> >
> > $$
> > \Pr\_{(x,y)\sim \mathcal{D}}\bigl[\mathcal{M}\_L(x)\neq y \wedge \mathcal{M}\_S(x)=y\bigr]\le \delta,
> > $$
> >
> > with $\delta \ll 1$. This “almost-always” dominance, supported by scaling-law trends (Kaplan et al., 2020), implies that deferring from $\mathcal{M}_S$ to $\mathcal{M}_L$ cannot hurt accuracy in expectation, while still allowing rare counter-examples where the small model outperforms the large model. Note that we empirically observe $\delta=0$ across all tasks considered in this work, meaning that $\mathcal{M}_L$ strictly dominates $\mathcal{M}_S$. See extended discussion in Appendix B.5 for details on this assumption.
>
> **Stating this assumption explicitly helps clarify the scope of our setup, ensures that the deferral oracle remains a valid performance ceiling, and provides a principled middle ground reconciling our experimental findings with the reviewer’s observation**. As stated in the assumption paragraph, we have also included an additional appendix section B.5 where we include our insights for this discussion with the reviewer as well as our justification for this assumption.
>
> ---
>
> **We hope that our response has clarified the reviewer’s concern and continue to be available for further clarification requests.**

---

> > ### Comment · Reviewer_ri54 · 2025-08-04
> >
> > Thank you for your this input which I agree with. I will update my review accordingly.

---

> > > ### Author Response · Authors · 2025-08-05
> > > **Thank you!**
> > >
> > > We are happy to hear that our most recent clarifications have addressed the reviewer's remaining concern and we thank them for updating their score!

---

### Official Review · Reviewer_5Nnu · 2025-07-05

**Clarity:** 3
**Significance:** 2
**Originality:** 2
**Rating:** 5
**Confidence:** 2

**Summary:**

This paper suggests a loss function, "Gatekeeper," for calibrating smaller models, and uses it to fine-tune small models for use in model cascades.

**Questions:**

- The GATEKEEPER loss looks a bit odd as it ignores the entropy of $y_i$. For example if for some example $x$ there are two equally likely options $y,y'$, then even the Bayes optimal predictor will incur $\mathcal{L}_\text{incorr}$. Could the authors provide some intuition for why such a loss function would still make sense in this setting?
- I have the same question for the gating function. In settings where $M_L$ would be uncertain, the gating mechanism will always defer the input to the larger model.
- More generally, the GATEKEEPER loss only uses the predicted and true labels and does not even require query access to $M_L$. Could the fine-tuning process be improved given access to the logits of $M_L$ by trying to predict when the output distributions of $M_S$ and $M_L$ will differ, rather than quantifying the uncertainty of $M_S$?
- Could the authors clarify the connection between Gatekeeper and calibration? While I agree that Gatekeeper will push the model towards the uniform distribution whenever it is unsure, this is not quite the same as calibration.

**Ethical Concerns:**

["NO or VERY MINOR ethics concerns only"]

**Final Justification:**

I originally recommended acceptance and decided to keep my score after the discussion phase.

**Limitations:**

yes

**Paper Formatting Concerns:**

I did not notice any major formatting issues in this paper.

**Quality:**

2

**Strengths And Weaknesses:**

Strengths:
- The paper is well written and easy to follow
- The proposed method is simple and only requires fine-tuning the small model $M_S$

Weaknesses:
- I don't fully understand the connection between Gatekeeper and calibration – see the questions below
- The proposed method does not leverage the query access to the large model $M_L$ for the fine-tuning, and only uses it during inference, which suggests further improvements could be made if the logits for $M_L$ are known on the training set.

---

> ### Author Rebuttal · Authors · 2025-07-30
>
> We thank the reviewer for their thoughtful evaluation and for **recognizing the practical relevance of our framework**. Their comments raise important points about the **assumptions behind the Gatekeeper loss, the gating behavior under uncertainty, and its connection to calibration**. We address each concern below, clarifying the loss design and discussing possible extensions. For all questions, we are happy to add extended sections to the appendix.
>
> ---
>
> ``1) Does Gatekeeper still make sense if we allow for examples with high inherent label noise?``
>
> ``2) Same question for the gating function.``
>
> We acknowledge that, when the predictive distribution $p(y|x)$ for a sample $x$ is inherently ambiguous—e.g. two labels $y,y^{\prime}$ are equally plausible—the Bayes-optimal predictor will place mass on both classes, and our KL-to-uniform term $\mathcal{L}_{\text{incorr}}$ will be triggered whenever the small model’s top-1 guess happens to miss the sampled ground-truth. This behaviour is in fact desirable for deferral: **the loss encourages the small model to *express* its uncertainty by flattening the output distribution whenever it is not confident**, thereby raising the probability that the subsequent gating function will defer the input. Conceptually, the system partitions samples into three mutually exclusive cases:
>
> - (i): $g(x)\ge\tau \ \land\  \hat{y}_{S}=y$
> - (ii): $g(x)<\tau \ \land\  \hat{y}_{L}=y$
> - (iii): $g(x)<\tau \ \land\  \hat{y}_{L}\neq y$
>
> corresponding to the small model answering correctly, the large model rescuing the mistake, and an irreducible error that neither model can solve. **Optimizing $\mathcal{L}_{\text{incorr}}$ improves the *boundary* between (i) and (ii)**, which is exactly the region that matters for useful deferral, while leaving case (iii) unaffected. Alternative objectives that try to anticipate the large model’s internal confidence would **collapse cases (ii) and (iii) into a single target but would require the small model to emulate the large model’s latent beliefs**—an infeasible endeavour in most practical scenarios and a key difference between *cascading* and *routing* strategies. Routing methods typically decide which model to send a query to (potentially partially querying any of the candidate models for determining the final routing decision itself), whereas cascading typically only uses signals produced by the small model. Finally, note that **our three-case breakdown explicitly accounts for inputs on which the large model itself remains uncertain (case (iii))**. In such instances the gate will indeed route the sample to $\mathcal{M}_L$, but this does not hurt joint accuracy—neither model can answer correctly—and only incurs an extra compute cost, which practitioners may cap by choosing a higher deferral threshold $\tau$ if desired
>
> ---
>
> ``3) Gatekeeper only uses the predicted and true labels and does not even require query access to M_L.``
>
> We agree that, **when query access to $\mathcal{M}_L$ is available, its logits can provide a complementary training signal—yet a key strength of Gatekeeper is that it *does not depend* on this access**, making it broadly deployable. Importantly, our training objective is easily augmented. One candidate solution would be to simply add an alignment term $\mathcal{L}_A = \frac{1}{N}\sum_i KL(p^{(S)}_i | | p^{(L)}_i)$ to our loss
>
> $$\mathcal{L}_\text{GK+} = \mathcal{L} + \beta \mathcal{L}_A$$
>
> where $p^{(S)}_i$ and $p^{(L)}_i$ denote the small- and large-model posteriors for sample $i$, and $\beta$ controls how aggressively we penalize divergences in their predictions. This term encourages $\mathcal{M}_S$ to *anticipate* situations in which its distribution would differ from $\mathcal{M}_L$, thereby sharpening the deferral signal when such discrepancies are likely.
>
> ---
>
> ``4) Connection between Gatekeeper and calibration.``
>
>
> Gatekeeper is *purpose-built for deferral*: the KL term deliberately flattens the softmax on examples the student misclassifies, **inducing *structured underconfidence* on precisely those inputs that should be routed to the large model**, while the cross-entropy term leaves already-correct examples confident. This selective re-weighting has two key effects:
> - **It “unlearns” brittle patterns on hard inputs**, freeing capacity to sharpen predictions on easy ones.
> - **It creates a much clearer separation between the confidence scores of correct and incorrect predictions**. Although this yields a model that is globally underconfident, it improves the *local calibration* that matters for risk–deferral: conditional on a confidence threshold, the realized accuracy tracks the nominal threshold far more closely, enabling a near-optimal deferral curve without any additional routing head.
>
> Empirically, we observe lower overconfidence on wrongly classified examples and substantially higher deferral performance than both the untuned baseline and prior cascading methods, confirming that this targeted form of calibration is precisely what a small local model needs to decide when to defer.
>
> ---
>
> **We hope that our rebuttal has clarified the reviewer’s questions and concerns and that the reviewer considers raising their individual score(s). We are happy to further engage with them during the discussion period should further questions arise!**

---

> > ### Comment · Reviewer_5Nnu · 2025-08-04
> >
> > Thank you for the clarifications. I have decided to keep my score.

---

> > > ### Author Response · Authors · 2025-08-04
> > > **Thank you!**
> > >
> > > We thank the reviewer for considering our rebuttal and are happy to hear about their continued strong support of our work!

---

### Comment · Area_Chair_mhQy · 2025-08-09
**Author-reviewer discussion**

Dear reviewers,

Thank you for your efforts and participating in discussion with the authors. We have a few hours remaining for author-reviewer discussion. If not already arrived at a conclusion in your discussions, can we please finalize now?

Dear aLX9: can you please indicate your conclusion based on the author response?

Thanks,
AC

---

### Decision · Program_Chairs · 2025-09-17

**Decision:**

Accept (poster)

**Comment:**

The submission proposes a novel gatekeeper objective for a smaller model to decide when to route harder examples to a larger model, to achieve a balance between accuracy and computational efficiency. 5Nnu questions the intuition of the proposed gatekeeper loss and its motivation relative to calibration, both of which are addressed satisfactorily by the author response. ri54 raises the concern that there might be conditions where M_L makes errors where M_S does not, to which the authors respond by diluting the dominance assumption to be that M_L dominates M_s with high probability. Both the reviewer and AC agree with this resolution. aLX9 raises limitations which might be addressed by a three-way decision across local accept, local reject and route to remote, which the authors analyze in the feedback but consider difficult to implement, which is convincing. aLX9 also requires discussion with respect to speculative decoding and early exiting, which are adequately included in the author response. Eqgu questions the contributions given existing literature on calibration, which is answered sufficiently well in the response and acknowledged as such by the reviewer. Overall, the AC agrees with the reviewer majority that the submission proposes an interesting method, with thorough analysis and convincing experiments. Thus, the submission is recommended for acceptance. The authors are encouraged to include discussions on limitations and clarifications on intuition for the proposed objective in the final version.